# Text-Trained LLMs Can Zero-Shot Extrapolate PDE Dynamics

## Abstract

Large language models (LLMs) have demonstrated emergent in-context learning (ICL) capabilities across a range of tasks, including zero-shot time-series forecasting. We show that text-trained foundation models can accurately extrapolate spatiotemporal dynamics from discretized partial differential equation (PDE) solutions without fine-tuning or natural language prompting. Predictive accuracy improves with longer temporal contexts but degrades at finer spatial discretizations. In multi-step rollouts, where the model recursively predicts future spatial states over multiple time steps, errors grow algebraically with the time horizon, reminiscent of global error accumulation in classical finite-difference solvers. We interpret these trends as in-context neural scaling laws, where prediction quality varies predictably with both context length and output length. To better understand how LLMs are able to internally process PDE solutions so as to accurately roll them out, we analyze token-level output distributions and uncover a consistent ICL progression: beginning with syntactic pattern imitation, transitioning through an exploratory high-entropy phase, and culminating in confident, numerically grounded predictions.

## 1 Introduction

Large language models (LLMs) exhibit an emergent ability known as in-context learning (ICL) (Brown et al., 2020; Dong et al., 2022; Zhao et al., 2025), in which the model is conditioned on a sequence of examples and/or task instructions provided in the input and learns to generate appropriate outputs for new instances—without any parameter updates or additional training on task-specific data. In the zero-shot setting, LLMs are given only a task description and/or a serialized input and are expected to generalize purely from the prompt.

While ICL was initially observed in linguistic tasks, it has since been demonstrated in domains involving mathematical reasoning (Wei et al., 2022a;b; Akyürek et al., 2023; Garg et al., 2022). Recent work shows that LLMs such as GPT-3 (Brown et al., 2020) and Llama-2 (Touvron et al., 2023) can, in the zero-shot setting, forecast time series (Gruver et al., 2023; Jin et al., 2024), infer governing principles of dynamical systems (Liu et al., 2024), and perform regression and density estimation (Requeima et al., 2024; Liu et al., 2025). From a theoretical perspective, in-context scaling laws have been analyzed by modeling LLM inference as a finite-state Markov chain, yielding analytical results for Markov-chain–generated inputs (Zekri et al., 2024), and by developing theoretical explanations of ICL scaling behavior when LLMs learn Hidden Markov Models (Dai et al., 2025).

We demonstrate that pretrained LLMs, such as Llama-3 (Grattafiori et al., 2024), Phi-4 (Abdin et al., 2024), and SmolLM3 (Hugging Face, 2025), possess an additional zero-shot ICL capability: the ability to continue the dynamics of partial differential equations (PDEs) directly from serialized solution data (see Section 3). Our focus is on time-dependent PDEs, whose solutions often exhibit multi-dimensional correlations, long-range dependencies, and stiff nonlinear dynamics (Evans, 2010; Haberman, 2013). We adopt the following setup: representing spatiotemporal data as delimited sequences of real numbers and feeding them directly into an LLM, without any fine-tuning or natural-language prompting. The model generates token sequences autoregressively, effectively learning to infer both spatial structure and temporal dynamics from in-context information alone (see Figure 1). We emphasize that we *do not* propose to employ LLMs as a new kind of PDE solver. Instead, we study their ICL behavior in continuing the spatiotemporal dynamics of PDEs, as a lens

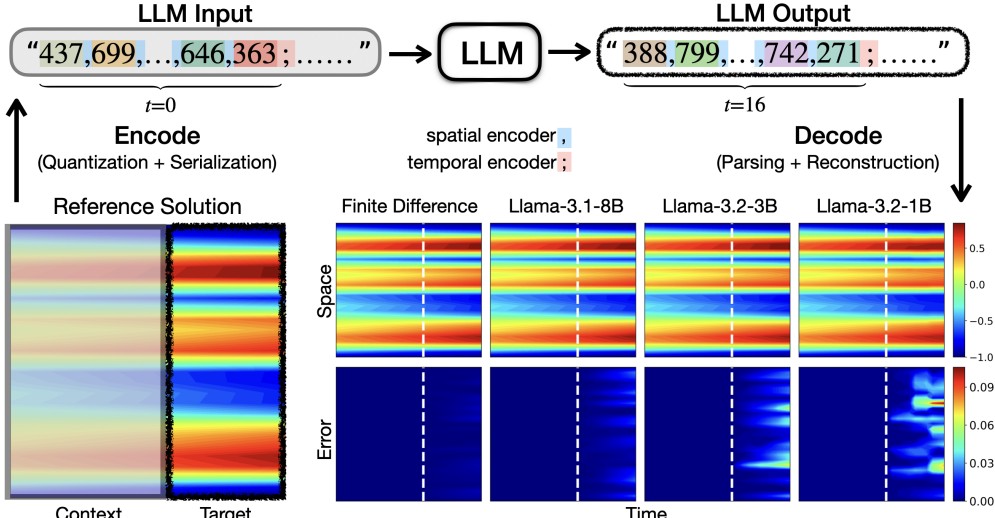

Figure 1: Zero-shot PDE extrapolation workflow with LLMs. A reference PDE solution to the Allen–Cahn equation is discretized over space and time, quantized to 3-digit integers, and serialized into a token sequence with spatial and temporal delimiters. Each value and delimiter is mapped to a token. The LLM autoregressively generates future tokens from past context without fine-tuning or natural language prompting. The generated tokens are parsed and reconstructed into floating-point solutions. LLM-predicted rollouts and absolute errors are compared against a numerical solver.

to investigate the inductive biases and numerical priors that emerge from large-scale pretraining. Anonymized code for reproducing all experiments is available in the supplementary material.

**Main Contributions.**

1) We demonstrate that pretrained LLMs exhibit robust zero-shot predictive capabilities on discretized PDE solutions with random initial conditions without fine-tuning or natural language prompting.

2) We identify in-context scaling laws for PDE-based spatiotemporal continuation with respect to temporal context length, spatial discretization, and rollout horizon, revealing behaviors analogous to truncation errors in classical numerical analysis.

3) We analyze token-level predictive entropy and uncover a consistent three-stage progression in ICL behavior during spatiotemporal PDE continuation.

## 2 BACKGROUND

Recent work at the intersection of LLMs and PDEs mainly follows two directions: (i) using LLMs as assistants in scientific modeling pipelines, and (ii) leveraging LLMs as direct PDE solvers. We briefly review representative examples from each line of research.

**LLMs as Assistants in PDE Pipelines.** Jiang et al. (2025) evaluate LLMs on tasks such as implementing numerical solvers and constructing scientific machine learning pipelines. Li et al. (2025) introduce CodePDE, a framework that formulates PDE solving as code generation. Soroco et al. (2025) propose PDE-Controller, which enables LLMs to convert informal natural language instructions into formal specifications for PDE control. Lorsung & Farimani (2024) leverage LLMs to integrate prior knowledge to improve PDE surrogate models. Zhou et al. (2025) present Unisolver, a neural PDE solver conditioned on symbolic PDE embeddings produced by LLMs.

**LLMs as PDE Solvers.** The Universal Physics Solver (UPS) (Shen et al., 2024) adapts pretrained LLMs to learn unified neural solvers for time-dependent PDEs. ICON-LM (Yang et al., 2025) fine-tunes LLMs for in-context operator learning. These methods typically involve custom architectures or task-specific training procedures tailored for PDE solving.

In contrast to prior work that adapts or fine-tunes LLMs for PDE solving, we investigate a zero-shot setting using pretrained LLMs, rather than proposing a new LLM architecture. We use this setup to study the numerical reasoning and inductive biases that emerge during the pretraining of LLMs, which are trained on textual data such as natural language and code (i.e., we do not consider multimodal foundation models).

## 3 METHODOLOGY

We extend the tokenization framework introduced by Gruver et al. (2023), which serializes one-dimensional time series as comma-separated numeric strings (e.g., "153, 412, ..., 807") for use with LLMs. Our approach, illustrated in Figure 1, generalizes this to time-dependent PDEs by converting discretized spatiotemporal solutions into structured 1D sequences. To encode both spatial and temporal structure, we introduce a two-delimiter format that separates spatial points and time steps. This representation preserves the underlying dynamics and enables interpretable analysis of how LLMs extrapolate higher-dimensional behavior.

**Grid Sampling.** We begin with a PDE solution $u(x, t)$ evaluated on a regular Cartesian spatiotemporal grid $\{u(x_i, t_j)\}_{i=1,j=0}^{N_{\mathrm{X}}, N_{\mathrm{T}}}$, represented as a matrix $\boldsymbol{U} \in \mathbb{R}^{N_{\mathrm{X}} \times (N_{\mathrm{T}}+1)}$. Each column $\boldsymbol{U}_{:,j}$ corresponds to the spatial state at time $t_j$.

**Quantization.** We apply linear quantization to map the continuous range $[u_{\min}, u_{\max}]$ to a fixed integer set[1] $\mathcal{Z} = \{150, 151, \ldots, 850\} \subseteq \mathbb{Z}$, yielding a quantized matrix $\boldsymbol{Q} \in \mathcal{Z}^{N_{\mathrm{X}} \times (N_{\mathrm{T}}+1)}$. This step introduces an underlying quantization error, computed as the difference between the original floating-point values $u(x_i, t_j)$ and their concomitant linearly reconstructed counterparts $\tilde{u}(x_i, t_j)$, obtained from quantized tokens (see Appendix A.1 for reconstruction details). We refer to this as the *quantization floor* in subsequent experiments.[2]

**Serialization.** Each time slice $\boldsymbol{Q}_{:,j}$ is serialized into a comma-separated string of 3-digit integers. Temporal evolution is encoded as a sequence of these strings, delimited by semicolons:

$$\text{“}\underbrace{Q_{1,j}, Q_{2,j}, \ldots, Q_{N_{\mathrm{X}},j}}_{\approx \boldsymbol{U}_{:,j}} ; \underbrace{Q_{1,j+1}, \ldots, Q_{N_{\mathrm{X}},j+1}}_{\approx \boldsymbol{U}_{:,j+1}} ; \ldots \text{”}$$

We adopt commas to delimit spatial entries, extending the CSV-style format of Gruver et al. (2023) to our setting. To represent the additional temporal dimension in a 2D spatiotemporal matrix, we introduce semicolons to mark time-step boundaries. This enhances parsability and aligns with familiar conventions: semicolons denote row breaks in MATLAB arrays and signal the end of statements in many programming languages (e.g., C, C++, Java). Linguistically, the semicolon also marks a stronger pause than a comma, reinforcing its role as a clear separator between time steps.

**Tokenizer Compatibility.** We adopt tokenizer configurations (e.g., GPT-4 (Achiam et al., 2023), Llama-3) in which each 3-digit value (000–999) and each delimiter (, and ;) maps to a single token. This one-to-one mapping directly aligns token positions with grid values in discretized PDE solutions, enabling efficient error computation and, importantly, direct estimation of predictive uncertainty at each location from the model's softmax outputs. In contrast, some models, such as Gemma 3 (Team et al., 2025), tokenize numeric values at different granularities (e.g., each digit as one token). Spatial value probabilities can still be recovered using hierarchical softmax methods in such models (Gruver et al., 2023; Liu et al., 2024), but at a higher computational cost. We therefore focus on LLMs with 3-digit tokenizers, while our serialization remains compatible with other tokenization schemes.

**LLM Inference.** The serialized sequence is passed to LLMs without fine-tuning or any natural language prompting. Tokens are generated autoregressively using the default generation configuration, with each prediction conditioned on the preceding context. We consider two inference modes: one-step prediction, where the model is set to generate a single future time slice consisting of $2N_{\mathrm{X}}-1$ tokens ($N_{\mathrm{X}}$ value tokens and $N_{\mathrm{X}}-1$ separator tokens); and multi-step rollouts, which repeat one-step prediction recursively, appending a semicolon after each time slice to indicate temporal progression. At each token position, we record both the model's output token (for prediction) and its full softmax

---

[1]000–149 and 851–999 are reserved; outputs in these ranges flag explicit out-of-distribution events.

[2]Reducible by enlarging $\mathcal{Z}$ beyond its 3-digit representation, at the cost of more tokens per value.

distribution (for uncertainty analysis). The generated sequence is parsed by splitting on semicolons to segment time and commas to recover spatial locations.[3]

## 4 EXPERIMENTS AND ANALYSIS

This section investigates the ability of state-of-the-art open-weight foundation LLMs to continue the spatiotemporal dynamics of PDEs. We focus our main analysis on the Allen–Cahn equation (Allen & Cahn, 1979), a nonlinear PDE modeling phase separation in multi-component metal alloy systems. To assess generality, we additionally evaluate the Fisher–KPP equation (Fisher, 1937; Kolmogorov et al., 1937), another nonlinear PDE modeling population growth and diffusion, together with two representative linear PDEs: the heat equation (a parabolic diffusion model) and the wave equation (a hyperbolic wave propagation model) (Evans, 2010). Across these families, with random initial conditions and varying boundary condition types, we observe qualitatively consistent ICL behavior. Remarkably, LLM rollouts also approximately conserve total thermal energy in the heat equation under Neumann boundaries, indicating that zero-shot ICL captures not only spatiotemporal dynamics but also structural invariants of PDEs. Full results and discussion of these additional PDE experiments are provided in Appendix A.3.

**PDE Setup.** By coupling reaction and diffusion dynamics, the Allen–Cahn equation induces strong interactions across space and time, making it a challenging yet physically interpretable testbed for assessing whether LLMs capture genuinely spatiotemporal structure rather than merely extrapolating along a single dimension. The system is defined on the interval $[-1, 1]$ with Dirichlet boundary conditions $u(-1, t) = u(1, t) = -1$ and random initial conditions $u(x, 0) = u_0(x)$ (details of initial condition generation in Appendix A.1). Explicitly, the Allen–Cahn PDE is:

$$\partial_t u = \epsilon^2 \, \partial_{xx} u - f(u), \quad x \in [-1, 1], \quad 0 \le t \le T.$$

Here, $\partial_t$ and $\partial_{xx}$ denote the temporal and second-order spatial derivatives, respectively. Adopting standard parameter choices (Raissi et al., 2019; Tang & Yang, 2016), we set the diffusion coefficient to $\epsilon^2 = 0.001$ and use a double-well potential $f(u) = 2(u^3 - u)$ for the nonlinear reaction term. The solution is evaluated on a uniform spatiotemporal grid $\{u(x_i, t_j)\}_{i=1, j=0}^{N_X, N_T}$, where $\{x_i\}_{i=1}^{N_X}$ are $N_X$ evenly spaced interior points and $\{t_j\}_{j=0}^{N_T}$ are $N_T + 1$ evenly spaced time levels, with $T = 0.5$.

**Numerical Benchmarks.** To contextualize the predictive structure learned by LLMs, we compare their outputs to two classical finite difference methods: the forward time, centered space (FTCS) scheme, which is fully explicit, and an implicit-explicit (IMEX) scheme that treats diffusion implicitly and the nonlinear reaction term explicitly (Smith, 1985). In contrast to developing new PDE solvers, our focus is on analyzing how pretrained foundation LLMs extrapolate PDE-based spatiotemporal dynamics in-context, without fine-tuning. Notably, their predictions can achieve surprising accuracy relative to standard numerical benchmarks, generalizing across varied initial conditions and discretizations. These results position PDEs as effective vehicles for examining the inductive biases and generalization behaviors of LLMs.

**Analysis Task Overview.** We analyze how LLMs generalize in autoregressively continuing PDE dynamics through two analytical lenses: (i) a truncation-error perspective, motivated by local and global error analysis in numerical PDEs (LeVeque, 2007; Larsson & Thomée, 2003), examining how prediction accuracy depends on discretization, rollout horizon, and model size; and (ii) a "systems-level" perspective, investigating how LLMs internalize and extrapolate PDE structure during ICL via entropy-based uncertainty measures. Section 4.1 analyzes one-step prediction, where accuracy improves with longer temporal context but degrades with finer spatial discretization. Section 4.2 analyzes multi-step rollouts, showing algebraic error growth with rollout horizon, analogous to global error accumulation in numerical solvers. Section 4.3 analyzes token-level uncertainty, revealing a consistent three-stage ICL progression: syntax mimicry, high-entropy exploration, and confident prediction. Prediction quality is evaluated using the Root Mean Square Error (RMSE), computed by

---

[3]Models rapidly internalize the delimiter structure. Even with minimal context (e.g., one time slice with five spatial points), comma delimiters are consistently generated. Malformed outputs are exceedingly rare, and parsing remains robust across multi-step rollouts. See Section 4.3 for quantitative results.

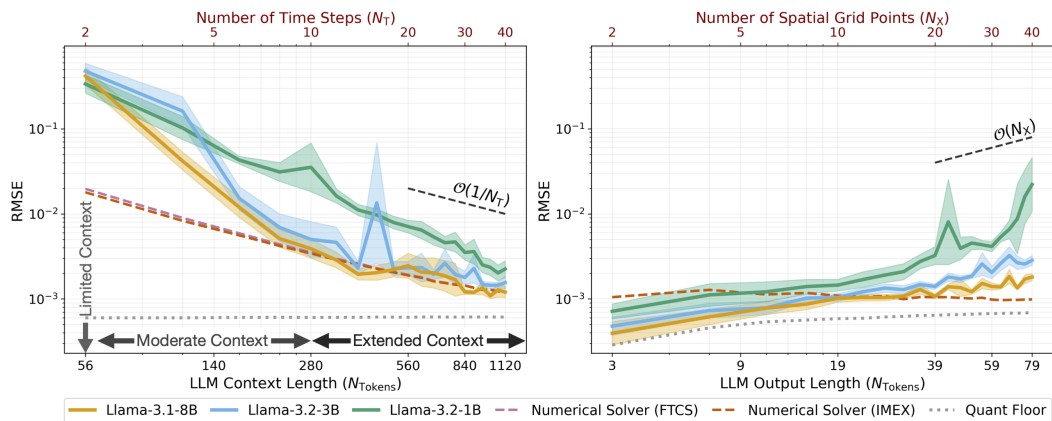

Figure 2: In-context error scaling with temporal discretization (**left**) and spatial discretization (**right**). The top axes show $N_\mathrm{T}$ and $N_\mathrm{X}$, while the bottom axes show the equivalent LLM context and output lengths $N_\mathrm{Tokens}$, respectively. RMSE decreases with longer context, converging in the extended-context regime, toward the local truncation behavior of first-order-in-time solvers (FTCS, IMEX). In contrast, errors grow with output length, following a capacity-dependent generalization trend. Shaded regions show 95% confidence intervals over 50 random initial conditions. The gray dotted line indicates the unavoidable quantization error floor defined in Section 3.

aggregating errors over all spatial grid points $\{x_i\}_{i=1}^{N_\mathrm{X}}$ per time step $\{t_j\}_{j=1}^{N_\mathrm{T}}$:

$$
\mathrm{RMSE}_j = \left( \frac{1}{N_\mathrm{X}} \sum_{i=1}^{N_\mathrm{X}} \left( \tilde{u}(x_i, t_j) - \hat{u}(x_i, t_j) \right)^2 \right)^{1/2}.
$$

RMSE measures overall predictive accuracy. Results using the Maximum Absolute Error, capturing worst-case deviation, show qualitatively similar trends (Appendix A.2). To ensure that our error metrics capture physically meaningful discrepancies—rather than differences between raw 3-digit quantized integer tokens—we compare each predicted solution $\hat{u}(x_i, t_j)$ against a floating-point reference solution $\tilde{u}(x_i, t_j)$. The reference $\tilde{u}(x_i, t_j)$ is computed on a highly refined finite-difference grid and passed through the same quantization–reconstruction pipeline (see Section 3) to ensure a consistent evaluation basis. LLM predictions, generated as token sequences, are likewise reconstructed into floating-point form before error evaluation. Since classical solvers operate on floating-point data, we apply the same quantization–reconstruction process to their initial conditions, ensuring that both LLM-based and classical methods operate on inputs of matched precision. Errors are then averaged over multiple random initial conditions, with further details in Appendix A.2.

**Model Setup.** Our primary results focus on *base* pretrained models from the Meta Llama-3 family: Llama-3.1-8B, Llama-3.2-3B, and Llama-3.2-1B. We also evaluate their instruction-tuned counterparts optimized for dialogue in Appendix A.4, which display qualitatively similar trends. For broader comparison, we include Microsoft Phi-4 and Hugging Face SmolLM3, which exhibit similar behaviors with differences mainly in error magnitude, in Appendix A.5.

### 4.1 ONE-STEP PREDICTION

We evaluate one-step prediction error under the following setup: given a discretized PDE solution from the initial condition up to one step before the final time, $\{u(x_i, t_j)\}_{i=1, j=0}^{N_\mathrm{X}, N_\mathrm{T}-1}$, the model predicts the terminal step $\{u(x_i, t_{N_\mathrm{T}})\}_{i=1}^{N_\mathrm{X}}$ across all spatial grid points. This prediction target is employed in order to isolate the effects of temporal and spatial discretization on accuracy. To ensure the task remains non-trivial, we select discretizations where solutions vary significantly between time steps, avoiding degenerate cases in which the model could succeed through trivial pattern repetition. Appendix A.7 provides empirical validation of this design choice.

**Longer Context Length Improves Prediction Accuracy.** We analyze how one-step prediction accuracy varies with input context length, measured by the number of observed time steps $N_\mathrm{T}$

provided to the LLM during ICL. Fixing the spatial discretization at $N_\mathrm{X} = 14$, we vary $N_\mathrm{T}$ from 2 to 40, spanning minimal to moderately informative temporal contexts. As $N_\mathrm{T}$ increases, the LLM receives a longer input sequence, and RMSE decreases consistently (Figure 2, left). On a log–log scale, the error curves exhibit approximate $\mathcal{O}(1/N_\mathrm{T})$ decay rate, closely resembling the convergence behavior of first-order-in-time solvers such as FTCS and IMEX. This suggests that LLMs exhibit inductive biases analogous to local truncation error in classical numerical methods.

Closer examination of Figure 2 (left) reveals three distinct stages as the temporal context increases. The emergence and evolution of these stages—and their connection to prediction uncertainty—are further analyzed in Section 4.3. In the **limited-context** stage ($N_\mathrm{T} = 2$), LLMs exhibit substantially higher errors compared to classical solvers. This behavior arises from surface-level pattern imitation in the solution format, rather than learning the underlying dynamical structure (see Section 4.3). In the **moderate-context** stage ($2 < N_\mathrm{T} < 10$), errors decay more rapidly than those of standard numerical benchmarks such as FTCS and IMEX, suggesting that LLMs move beyond pattern imitation and begin to internalize aspects of the governing PDE dynamics. Finally, in the **extended-context** stage ($N_\mathrm{T} \geq 10$), error decay closely matches that of classical first-order solvers, indicating that LLMs are effectively leveraging the spatiotemporal structure in a numerically grounded way. In this stage, Llama-3.1-8B consistently matches, or in some cases exceeds, the accuracy of classical solvers. Overall, this reveals an empirical in-context scaling law: increasing the input context length consistently improves prediction accuracy, reflecting the LLM's increasing ability to internalize and extrapolate latent PDE dynamics at fixed spatial discretization.

**Longer Output Length Degrades Prediction Accuracy.** We analyze how prediction accuracy varies with the number of spatial discretization points $N_\mathrm{X}$. Since LLMs predict the solution at all spatial grid points for a given time step, increasing $N_\mathrm{X}$ directly results in a proportionally longer output sequence. To isolate this effect, we vary $N_\mathrm{X}$ across 20 evenly spaced values from 2 to 40, while fixing the temporal context at $N_\mathrm{T} = 50$. As shown in Figure 2 (right), RMSE grows consistently with $N_\mathrm{X}$, following an approximate $\mathcal{O}(N_\mathrm{X})$ scaling law on a log–log scale. The error growth is steepest for the smaller Llama-3.2-1B model, while the larger Llama-3.1-8B shows slower growth, indicating improved robustness to output length within the Llama-3 family. This behavior stands in sharp contrast to classical finite-difference solvers, where increasing spatial resolution typically does not raise error under a stable scheme. For LLMs, however, outputs are generated as flat autoregressive token sequences: larger $N_\mathrm{X}$ leads to longer, more complex outputs that must be generated without access to the underlying PDE, placing growing demands on the model's ICL capacity.

These findings reveal a second empirical in-context scaling law: finer spatial discretization produces longer outputs and degrades prediction accuracy under fixed input context. The effect scales strongly with model size within the Llama-3 family. Smaller models face more pronounced performance drops, indicating that limited ICL capacity constrains generalization at finer spatial discretizations.

## 4.2 MULTI-STEP ROLLOUTS

We examine LLMs' capacity to continue PDE solutions over multiple time steps based solely on in-context input. For a rollout of $N_\mathrm{T}$ time steps, we partition the serialized sequence into a context segment of $\lfloor \frac{2}{3} N_\mathrm{T} \rfloor$ steps and a prediction segment with the remainder. For $N_\mathrm{T} = 25$, this yields 16 context steps (including the initial condition), $\{u(x_i, t_j)\}_{i=1,j=0}^{N_\mathrm{X},15}$, which the LLM uses to autoregressively generate 10 prediction steps, $\{u(x_i, t_j)\}_{i=1,j=16}^{N_\mathrm{X},25}$, without access to intermediate ground truth. This 2:1 context-to-prediction ratio strikes a balance between providing sufficient context and posing a nontrivial extrapolation challenge. We assess model behavior via (i) representative rollouts from single random initial conditions, and (ii) average error trends over random initial conditions to quantify how prediction error accumulates over the prediction horizon.

**Qualitative Multi-Step Rollouts.** Figure 3 illustrates representative multi-step rollouts for the Allen–Cahn and wave equations, showing that LLMs can sustain coherent, qualitatively accurate predictions over a 10-step horizon. This is notable because the foundation models are not specialized PDE solvers and lack training-time exposure to the discretized PDE solutions; initial conditions are randomly sampled at inference time. The Llama-3.1-8B model closely tracks the evolution, capturing nonlinear reaction–diffusion dynamics and finite-speed wave propagation without collapsing to trivial behavior or diverging. When prompted with sufficiently long input context, the model can approximate and extrapolate PDE dynamics purely through autoregressive token-level inference. In

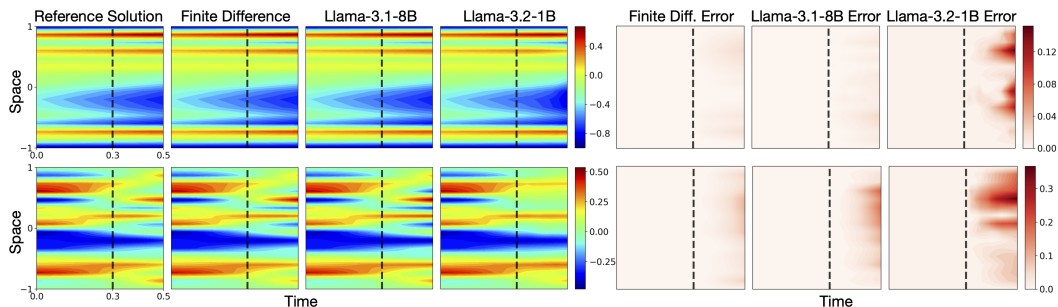

Figure 3: Multi-step prediction for randomly sampled initial conditions of two PDEs. The first row shows the Allen–Cahn equation, and the second shows the wave equation ($c = 0.3$; see Appendix A.3). In each case, to the left of the dashed line corresponds to the input context provided to the LLM, and to the right corresponds to a 10-step autoregressive continuation from a single generation of each model. Classical finite difference solvers (FTCS for Allen–Cahn, leapfrog for wave) solve the corresponding initial value problem using the final in-context time slice as the initial condition, and advance the solution for 10 steps using the same spatial and temporal discretization as the LLMs. The final three columns report pointwise absolute errors relative to the reference solution.

contrast, Llama-3.2-1B shows larger deviations and fails to preserve spatiotemporal structure over extended rollouts (see Appendix A.9 for analysis of error patterns and capacity limits). Visualizations for Llama-3.2-3B, additional numerical benchmarks, and further random initial conditions are provided in Appendix A.8.

**Quantitative Multi-Step Error Growth.** To characterize error growth over the prediction horizon, we repeat the multi-step rollout procedure for the Allen–Cahn equation across 20 randomly sampled initial conditions. Averaging over initializations reveals a consistent algebraic increase in RMSE with rollout length, as shown on a log–log scale in Figure 4. This resembles global error accumulation in classical finite-difference solvers operating under stable discretizations, where local truncation errors compound in a controlled manner over time. Crucially, none of the models exhibit divergent or unstable behavior across the 10-step horizon; error growth remains bounded. These findings underscore the capacity of LLMs to continue PDE dynamics via ICL across diverse initial conditions, sustaining coherent predictions over extended horizons—a fundamentally nontrivial task given only in-context information, without prompting or access to governing equations. Similar error-growth trends are observed for the wave equation and other PDEs, as detailed in Appendix A.3.

### 4.3 UNCERTAINTY EVOLUTION AND LEARNING STAGES IN IN-CONTEXT LEARNING

We now move beyond truncation-like error analysis to examine how LLMs internalize PDE dynamics through ICL. We focus on predictive uncertainty and generation behavior in the one-step prediction task introduced in Section 4.1.[4] We vary context length via temporal discretization ($N_{\mathrm{T}}$) and output length via spatial discretization ($N_{\mathrm{X}}$), and analyze how these factors shape the model's token-level uncertainty. Given input $\{u(x_i, t_j)\}_{i=1,j=0}^{N_{\mathrm{X}}, N_{\mathrm{T}}-1}$, the model predicts $\{u(x_i, t_{N_{\mathrm{T}}})\}_{i=1}^{N_{\mathrm{X}}}$. To quantify predictive uncertainty, we compute the Shannon entropy (Shannon, 1948) of the model's softmax distribution at each spatial value token and average across space:

$$\bar{H}(N_{\mathrm{T}}, N_{\mathrm{X}}) = -\frac{1}{N_{\mathrm{X}}} \sum_{i=1}^{N_{\mathrm{X}}} \sum_{y \in \mathcal{V}} p(y \mid x_i, N_{\mathrm{T}}) \log p(y \mid x_i, N_{\mathrm{T}}),$$

where $\mathcal{V}$ denotes the tokenizer's vocabulary, and $p(y \mid x_i, N_{\mathrm{T}})$ is the predicted probability of token $y$ at spatial location $x_i$, given $N_{\mathrm{T}}$ prior time steps in the serialized input. See Appendix A.2 for implementation details.

**Emergent Learning Stages Revealed by Entropy Evolution.** Figure 5A shows the evolution of mean spatial entropy, $\bar{H}$, as a function of $N_{\mathrm{T}}$. A distinct rise-and-fall pattern reveals three emergent

---

[4]Since the multi-step rollouts analyzed in Section 4.2 recursively apply the one-step prediction process, we defer a parallel analysis of uncertainty accumulation in that setting to Appendix A.10.

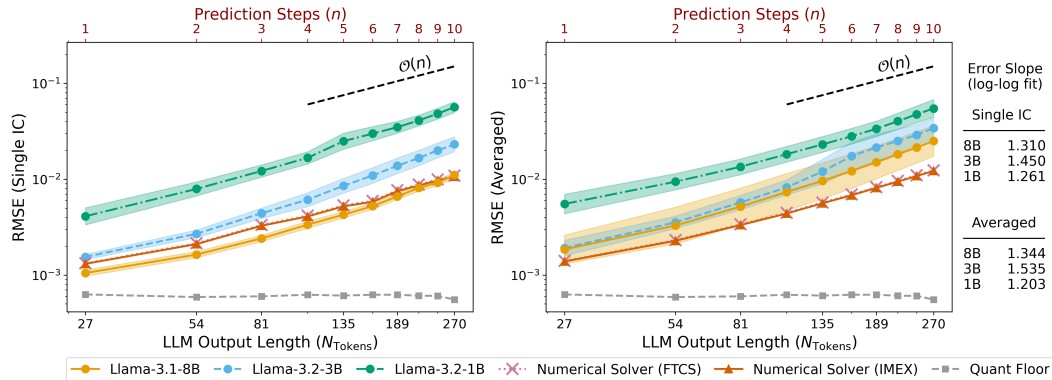

Figure 4: Multi-step rollout error trends. RMSE grows algebraically with prediction steps $n$ (top axis) and equivalent LLM output length $N_{\text{Tokens}}$ (bottom axis). **Left**: rollout from a single random initial condition (as in Figure 3). **Right**: average over 20 random initial conditions. Error growth rates are estimated via log–log fits and reported on the right. Shaded regions denote 95% confidence intervals (left: across 20 repeated LLM runs; right: across 20 initial conditions).

stages of ICL: an initial syntax-dominated stage, a transitional exploratory stage, and a final stage of consolidation and refinement:

1) **Syntax-Only** (limited context, e.g., $N_{\text{T}} = 2$; Figure 5C, first row): While mean spatial entropy $\bar{H}$ exhibits some variation across model sizes, prediction error remains consistently high. Separator tokens (e.g., commas) are predicted with near-perfect confidence (see Figure 5D). In contrast, spatial value tokens act as generic placeholders, with little correspondence to underlying PDE dynamics, yielding deterministic yet physically implausible predictions. This indicates that syntax is acquired before any meaningful understanding of the PDE dynamics emerges.

2) **Exploratory** (moderate context, e.g., $2 < N_{\text{T}} < 10$; Figure 5C, second row): Entropy reaches its peak across model sizes, indicating increased uncertainty and broader spatial token distributions. Meanwhile, prediction accuracy improves rapidly, and outputs begin to align with true PDE dynamics. This stage marks a transition from merely capturing surface-level syntax to beginning to internalize spatiotemporal dynamics.

3) **Consolidation** (extended context, e.g., $N_{\text{T}} \geq 10$; Figure 5C, third row): As context length increases further, $\bar{H}$ decreases, reflecting sharper and more confident spatial token distributions. Prediction accuracy continues to improve, though with less profound gains compared to the exploratory stage. The model's predictions increasingly reflect coherent and physically meaningful PDE dynamics.

These stages reveal a consistent ICL progression: 1) syntax acquisition, 2) exploratory numerical behavior, and 3) convergence to accurate predictions. This progression suggests that LLMs develop structured internal representations of PDE dynamics purely through in-context exposure, without explicit access to governing equations or language prompting.

**Uncertainty Growth with Output Length.** While the previous analysis focused on how input context length affects ICL and predictive uncertainty, we now examine how spatial discretization ($N_{\text{X}}$) affects prediction confidence. Fixing the temporal context at $N_{\text{T}} = 50$, we vary $N_{\text{X}}$ and compute the mean spatial entropy $\bar{H}$. As shown in Figure 5B, $\bar{H}$ increases steadily with larger $N_{\text{X}}$, reflecting growing uncertainty for longer spatial outputs. This trend mirrors the error scaling in Figure 2, where RMSE increases with $N_{\text{X}}$ under fixed input context. Notably, the smaller Llama-3.2-1B exhibits the steepest entropy growth, while the larger Llama-3.1-8B shows the slowest, indicating greater robustness to output length. These findings reveal a close empirical link between model uncertainty and prediction error: longer output sequences from finer spatial discretization lead to both higher entropy and reduced predictive accuracy. Within the Llama-3 family, larger models consistently maintain higher confidence and accuracy over extended output lengths. This suggests that model size plays a critical role in enabling accurate extrapolation of learned PDE dynamics across finer spatial discretizations—a capacity that smaller models fail to maintain.

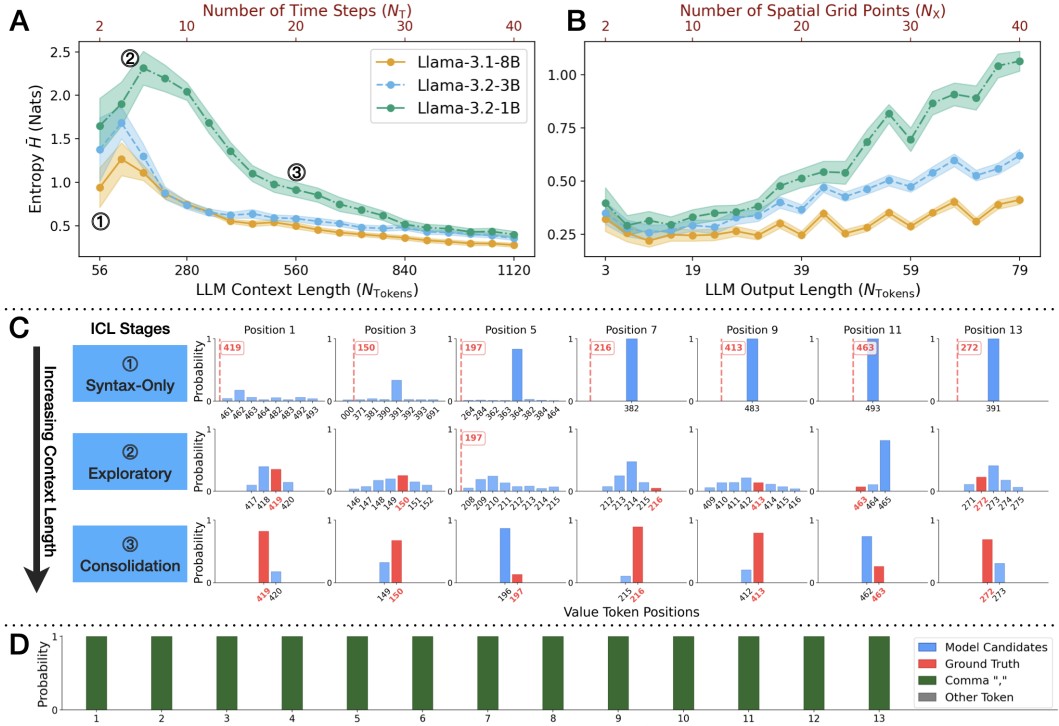

Figure 5: **(A–B)** Mean spatial entropy $\bar{H}$ vs. (A) temporal context length $N_T$ at fixed $N_X = 14$ and (B) output length $N_X$ at fixed $N_T = 50$. Shaded regions: 95% confidence intervals over 50 random initial conditions. **(C)** Token-level softmax distributions at three ICL stages: $N_T = 2$ (syntax-only), 5 (exploratory), and 20 (consolidation), extracted from Llama-3.1-8B for the same initial condition as the multi-step rollout example. Top 8 tokens (by probability) are shown per spatial position; only odd positions are displayed, with full results in Appendix A.11. **(D)** Softmax over separator tokens, invariant across stages.

## 5 CONCLUSION

We show that text-trained LLMs can extrapolate PDE dynamics in-context in a zero-shot setting, without any fine-tuning or natural language prompting. Their performance exhibits clear in-context scaling laws: accuracy improves with longer temporal context, degrades with finer spatial discretization, and error grows algebraically under multi-step rollouts. Entropy analysis further reveals a three-phase progression—from syntax imitation, to exploratory uncertainty, to stabilized predictions—highlighting emergent mechanisms underlying ICL. Together, these findings suggest that LLMs can internalize nontrivial aspects of PDE dynamics purely from in-context data, demonstrating emergent generalization capabilities in zero-shot inference.

**Limitations and Future Work.** Our study is restricted to time-dependent PDEs with real-valued solutions on regular domains. Extending this framework to stationary PDEs (e.g., Poisson), PDEs with complex-valued solutions (e.g., Schrödinger), higher-dimensional or irregular domains, and partially observed or noisy dynamics could reveal complementary behaviors. Beyond these extensions, a key challenge is to characterize the internal representations and compositional structures that support generalization over spatiotemporal dynamics in autoregressive token space. While we focus on demonstrating the surprising ICL abilities of text-trained LLMs for PDE dynamics continuation, an important open problem is to identify PDE regimes where LLMs outperform classical solvers, and to explore how symbolic priors or textual prompts (Xue & Salim, 2024; Requeima et al., 2024) could amplify these advantages.

DISCLOSURE OF LLM USAGE

The authors used LLMs solely for improving the wording of the manuscript text and the presentation of experiment code. No research ideas, analyses, or substantive content were generated by LLMs.

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

# A APPENDIX

## A.1 DETAILS ON EXPERIMENTAL SETUP AND INITIAL CONDITIONS

### A.1.1 QUANTIZATION AND RECONSTRUCTION IMPLEMENTATION

This appendix provides the implementation details for the linear quantization and reconstruction steps described in Section 3. Let $\{u(x_i, t_j)\}_{i=1,j=0}^{N_X, N_T}$ denote the floating-point PDE solution evaluated on a uniform spatiotemporal grid with $N_X$ interior spatial points $\{x_i\}_{i=1}^{N_X}$ and $N_T + 1$ discrete time steps $\{t_j\}_{j=0}^{N_T}$. Define

$$u_{\min} = \min_{i,j} u(x_i, t_j), \qquad u_{\max} = \max_{i,j} u(x_i, t_j).$$

We quantize into the integer set $\mathcal{Z} = \{150, 151, \ldots, 850\}$, resulting in a quantized matrix $\boldsymbol{Q} \in \mathcal{Z}^{N_X \times (N_T+1)}$.

**Quantization.** Each entry $u(x_i, t_j)$ is mapped to

$$Q_{i,j} = \begin{cases} 500, & \text{if } u_{\max} = u_{\min}, \\ \text{round}\Big(150 + (u(x_i, t_j) - u_{\min}) \dfrac{850 - 150}{u_{\max} - u_{\min}}\Big), & \text{otherwise}, \end{cases}$$

where $\text{round}(\cdot)$ denotes rounding to the nearest integer.

**Reconstruction.** To recover an approximate floating-point value $\tilde{u}(x_i, t_j)$ from $Q_{i,j}$, we apply a linear reconstruction map that approximates the original values:

$$\tilde{u}(x_i, t_j) = \begin{cases} u_{\min}, & \text{if } u_{\max} = u_{\min}, \\ u_{\min} + (Q_{i,j} - 150)\, \dfrac{u_{\max} - u_{\min}}{850 - 150}, & \text{otherwise.} \end{cases}$$

**Quantization Error Floor.** We report two per-time-step error metrics for the reconstructed solution:

$$\text{MaxAE}_j^Q = \max_{1 \leq i \leq N_X} \big|\tilde{u}(x_i, t_j) - u(x_i, t_j)\big|, \quad \text{RMSE}_j^Q = \left( \frac{1}{N_X} \sum_{i=1}^{N_X} \big(\tilde{u}(x_i, t_j) - u(x_i, t_j)\big)^2 \right)^{1/2}.$$

Each metric quantifies the unavoidable error floor introduced by first mapping the floating-point solution into the discrete integer set $\mathcal{Z} = \{150, \ldots, 850\}$, and then reconstructing it back to floating-point values via the linear approximation. We refer to these errors, $\{\text{MaxAE}_j^Q\}$ and $\{\text{RMSE}_j^Q\}$, collectively as the *quantization floor* under the corresponding metric.

**LLM Inference Setup.** For LLM inference described in Section 3, we consider only the token library $\mathcal{V}$ consisting of three-digit numbers (000–999) and the comma delimiter (,) that encodes spatial position. All other tokens outside $\mathcal{V}$ are masked, and sampling is renormalized over this set. This guarantees that every output corresponds to either a valid grid value entry or a spatial delimiter token, thereby preserving one-to-one alignment with the serialized PDE solution. For multi-step rollouts, temporal delimiters (semicolon) are inserted deterministically to mark time-step boundaries and appear in the LLM input context.

### A.1.2 Spline-Based Random Initial Condition Construction

In Section 4, we construct each random initial condition $u_0(x)$ by sampling independent values on a fixed grid and then fitting a $C^2$ interpolant via cubic splines, yielding a function that is $C^2$ on $[-L, L]$. This analytic procedure is computationally cheap and helps ensure that any variation in LLM prediction error across different spatial discretizations arises purely from discretization, not from changes in the underlying random sample (see Section 4.1).

**Coarse Grid and Sampling.** Let the 1D domain be $[-L, L]$, with Dirichlet boundary values

$$u(-L, 0) = u(L, 0) = u_{\text{BC}}.$$

Introduce a uniform grid of $N_X$ interior points and two boundary points:

$$x_i = -L + i\, \Delta x, \quad \Delta x = \frac{2L}{N_X + 1}, \quad i = 0, 1, \ldots, N_X + 1,$$

where $i = 0$ and $i = N_X + 1$ correspond to the boundaries. Draw interior values independently and identically from a uniform distribution on $[a, b]$,

$$u_i \sim \mathcal{U}[a, b], \quad i = 1, \ldots, N_X,$$

where $a < b$ are the lower and upper bounds. Then assemble the fixed-value vector

$$\boldsymbol{u}^{\text{fixed}} = \big[u_0,\, u_1,\, \ldots,\, u_{N_X},\, u_{N_X+1}\big] = \big[u_{\text{BC}},\, u_1,\, \ldots,\, u_{N_X},\, u_{\text{BC}}\big].$$

**Spline Interpolant.** Use SciPy's `CubicSpline` with default "not-a-knot" end conditions to fit

$$S(x) = \text{Spline}\big(\{x_i\}_{i=0}^{N_X+1},\, \boldsymbol{u}^{\text{fixed}}\big),$$

which yields a twice continuously differentiable function. Define the continuous initial condition

$$u_0(x) = S(x).$$

**Resampling at a New Grid Resolution.** To evaluate different spatial discretizations, choose a new cardinality of interior points $N_X^{\text{new}}$ and set

$$x_j^{\text{new}} = -L + j\, \Delta x^{\text{new}}, \quad \Delta x^{\text{new}} = \frac{2L}{N_X^{\text{new}} + 1}, \quad j = 0, 1, \ldots, N_X^{\text{new}} + 1.$$

Then define

$$u_j^{\text{new}} = \begin{cases} u_{\text{BC}}, & j \in \{0, N_{\text{X}}^{\text{new}} + 1\}, \\ S\big(x_j^{\text{new}}\big), & 1 \le j \le N_{\text{X}}^{\text{new}}. \end{cases}$$

The set $\{u_j^{\text{new}}\}$ provides the discrete initial data at the finer (or coarser) grid. By holding $S(x)$ fixed, this approach isolates the effect of grid spacing on one-step prediction error.

**Parameter Choices.** All experiments in Section 4 for the Allen–Cahn PDE are conducted with $N_{\text{X}} = 14, u_{\text{BC}} = -1, a = -0.5, b = 0.5$.

## A.2 EVALUATION METRICS AND REFERENCE SOLUTION SETUP

In this appendix, we detail the Monte Carlo procedure used to compute the evaluation metrics in Section 4. For completeness, we also report results using the Maximum Absolute Error (MaxAE), which captures the worst-case deviation:

$$\text{MaxAE}_j = \max_{1 \le i \le N_{\text{X}}} |\tilde{u}(x_i, t_j) - \hat{u}(x_i, t_j)| \,.$$

As shown in Figure 6, MaxAE exhibits qualitatively similar trends to the RMSE reported in the error analysis of Section 4. Algorithm 1 summarizes the Monte Carlo procedure used for computing evaluation metrics in the one-step prediction task. The same procedure extends naturally to the multi-step setting, where the metrics are computed identically at each predicted time slice.

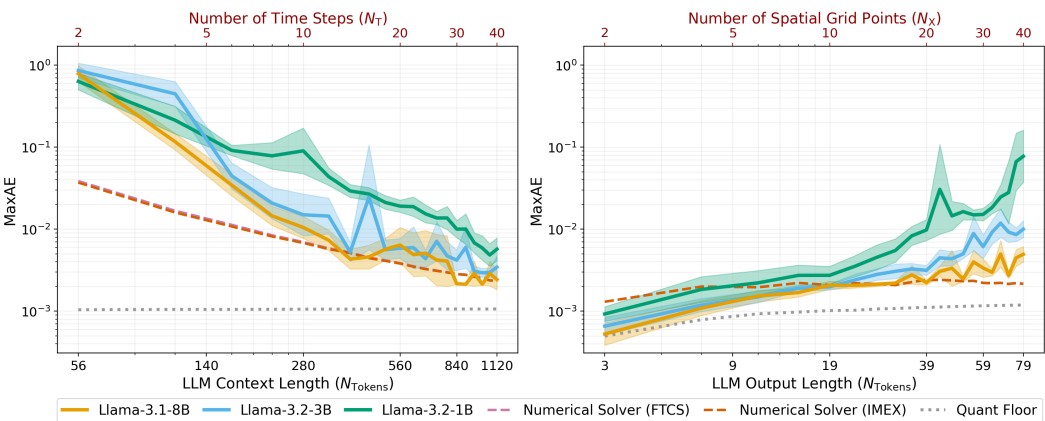

(a) One-step prediction vs. input context/output length.

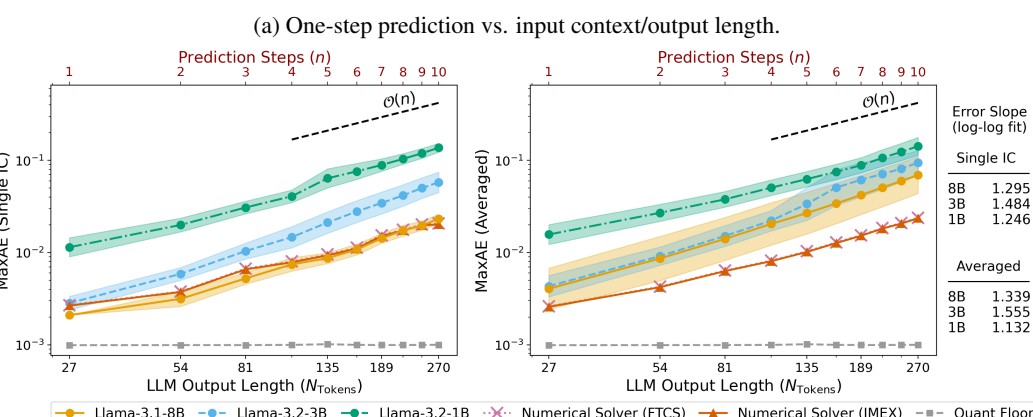

(b) Multi-step rollouts vs. prediction steps.

Figure 6: Prediction accuracy of Llama-3 models evaluated using the MaxAE metric, under the same experimental setup as in Sections 4.1 and 4.2.

---

**Algorithm 1** Metrics Calculation for Numerical Results

---

1: **Fixed Quantities**: Initial conditions $\{\tilde{u}_{0,m}\}_{m=1}^{M}$ and corresponding reference solutions $\{\tilde{u}_m(x,t)\}_{m=1}^{M}$ precomputed on a suitably highly refined finite-difference grid using appropriate schemes (FTCS for Allen–Cahn and Fisher–KPP; BTCS for heat; leapfrog for wave).

2: **Monte Carlo Trials**: For $m = 1, 2, \ldots, M$, run $A \in \{\text{LLM}, \text{Classical Solver}\}$ given $\{\tilde{u}_m(x_i, t_j)\}_{i=1, j=0}^{N_X, N_T - 1}$ and obtain prediction $\{\hat{u}_m^A(x_i, t_{N_T})\}_{i=1}^{N_X}$.

$$\text{MaxAE}_m^A = \max_{1 \leq i \leq N_X} \left| \tilde{u}_m(x_i, t_{N_T}) - \hat{u}_m^A(x_i, t_{N_T}) \right|, \qquad \text{(Maximum Absolute Error)}$$

$$\text{RMSE}_m^A = \left( \frac{1}{N_X} \sum_{i=1}^{N_X} \left( \tilde{u}_m(x_i, t_{N_T}) - \hat{u}_m^A(x_i, t_{N_T}) \right)^2 \right)^{1/2}. \qquad \text{(Root Mean Square Error)}$$

For $A = \text{LLM}$ only, compute:

$$\bar{H}_m^{\text{LLM}} = -\frac{1}{N_X} \sum_{i=1}^{N_X} \sum_{y \in \mathcal{V}} p(y \mid x_i, N_T) \log p(y \mid x_i, N_T), \qquad \text{(Mean Entropy)}$$

where $\mathcal{V}$ is the LLM's token vocabulary and $p_m(y \mid x_i, N_T)$ is the softmax probability for token $y$ at location $x_i$ in trial $m$.

3: **Error Metrics**: Mean error and corresponding 95% confidence interval on the log scale (to match error plots that span multiple orders of magnitude): For $E \in \{\text{MaxAE}, \text{RMSE}\}$,

$$E^A = \frac{1}{M} \sum_{m=1}^{M} E_m^A, \qquad \text{(Averaged Error Metric)}$$

$$\sigma_E^A = \left( \frac{1}{M-1} \sum_{m=1}^{M} (E_m^A - E^A)^2 \right)^{1/2}, \qquad \text{(Sample Standard Deviation)}$$

$$\log_{10}(\text{CI}_E^A) = \log_{10}(E^A) \pm t_{0.975, M-1} \cdot \frac{\sigma_E^A}{E^A \cdot \sqrt{M} \cdot \ln(10)}, \qquad \text{(95\% CI)}$$

where $t_{0.975, M-1}$ is the 97.5th percentile of the Student's t-distribution with $M-1$ degrees of freedom.

4: **Uncertainty Metrics** (LLM only): Mean entropy and corresponding 95% confidence interval on the regular scale (to match uncertainty plots):

$$\bar{H}^{\text{LLM}} = \frac{1}{M} \sum_{m=1}^{M} \bar{H}_m^{\text{LLM}}, \qquad \text{(Averaged Entropy)}$$

$$\sigma_{\bar{H}}^{\text{LLM}} = \left( \frac{1}{M-1} \sum_{m=1}^{M} (\bar{H}_m^{\text{LLM}} - \bar{H}^{\text{LLM}})^2 \right)^{1/2}, \qquad \text{(Sample Standard Deviation)}$$

$$\text{CI}_{\bar{H}}^{\text{LLM}} = \bar{H}^{\text{LLM}} \pm t_{0.975, M-1} \cdot \frac{\sigma_{\bar{H}}^{\text{LLM}}}{\sqrt{M}}. \qquad \text{(95\% CI)}$$

---

### A.3    Results on Additional PDEs

#### A.3.1    Results on Fisher–KPP, Heat, and Wave Equations

We extend the analysis from Section 4 to three additional PDEs: the Fisher–KPP equation, the heat equation, and the wave equation. All experiments (one-step prediction, multi-step rollout, and entropy-based uncertainty quantification) are performed under the same setup as in Section 4,[5] with homogeneous Dirichlet boundary conditions unless noted otherwise. For the wave equation, which is second-order in time, we additionally impose zero initial velocity, i.e., $\partial_t u(x, 0) = 0$, so that the dynamics are fully determined by the initial condition $u(x, 0)$. Explicitly, the governing equations are:

$$\begin{aligned}
\text{Fisher–KPP:} \quad & \partial_t u = D\, \partial_{xx} u + ru(1 - u), \\
\text{Heat:} \quad & \partial_t u = k\, \partial_{xx} u, \\
\text{Wave:} \quad & \partial_{tt} u = c^2\, \partial_{xx} u.
\end{aligned}$$

For the Fisher–KPP equation, we adopt commonly used parameter values with diffusion coefficient $D = 0.002$ and reaction rate $r = 1$ (Hasnain & Saqib, 2017; Needham et al., 2025). In the representative results (Figures 7–10), we present results with thermal diffusivity $k = 0.01$ and wave speed $c = 0.2$. In Figure 11, we further confirm that the qualitative scaling trends persist across a range of $k$ and $c$ values. For each PDE, we additionally include representative numerical benchmarks to contextualize LLM predictions: FTCS and IMEX for the Fisher–KPP equation, FTCS and BTCS (backward time, centered space) for the heat equation, and leapfrog and Crank–Nicolson for the wave equation (LeVeque, 2007). Beyond Dirichlet boundaries, we also study the heat equation under homogeneous Neumann boundary conditions, where total thermal energy conservation is the key structural property. Notably, LLM rollouts preserve this conservation law, suggesting that ICL can capture deeper invariants of PDE dynamics. Full details are provided in Appendix A.3.2.

As shown in Figures 7, 8, 9, and 10, the qualitative trends closely mirror those observed for the Allen–Cahn equation presented in Section 4. In the one-step prediction setting, accuracy improves systematically with longer temporal context while degrading at finer spatial discretizations. In the multi-step rollout setting, errors accumulate algebraically with the rollout horizon, resembling the global error growth of classical numerical solvers. Entropy-based analysis reveals a consistent three-stage progression in ICL behavior, and prediction uncertainty increases with longer spatial outputs.

Overall, these results demonstrate that the emergent in-context scaling laws and uncertainty dynamics observed for the Allen–Cahn equation persist across PDE families with markedly different physical behaviors: nonlinear growth–diffusion, heat diffusion, and wave propagation. The persistence of these patterns underscores the robustness and generality of LLM ICL on continuing spatiotemporal PDE dynamics, suggesting that foundation models possess inductive biases that allow them to internalize and extrapolate PDE dynamics.

#### A.3.2    Conservation Properties in the Heat Equation with Neumann Boundaries

**Motivation.** Beyond continuing spatiotemporal trajectories, an important question is whether LLMs internalize deeper invariants of PDE dynamics. The heat equation with homogeneous Neumann boundary conditions and no internal source term offers a natural test case: it models an insulated rod, where no heat can flow across the boundaries. In this setting, the total thermal energy is conserved for all time (Haberman, 2013). Remarkably, we find that LLM rollouts respect this conservation law more faithfully than coarse-grid finite difference solvers under the same setup, suggesting that ICL captures structural properties of the dynamics rather than performing naive extrapolation.

**Conservation Law.** The governing PDE is:

$$\partial_t u(x, t) = k\, \partial_{xx} u(x, t), \qquad \partial_x u(-L, t) = \partial_x u(L, t) = 0.$$

Define the total thermal energy:

$$\mathrm{E}(t) = \int_{-L}^{L} u(x, t)\, dx.$$

---

[5]For the Fisher–KPP equation, we set $a = 0.2, b = 0.8$ (instead of $a = -0.5, b = 0.5$), following Appendix A.1.2 so that the initial condition $u(x, 0)$ lies within $[0, 1]$, consistent with interpreting $u(x, t)$ as a population density.

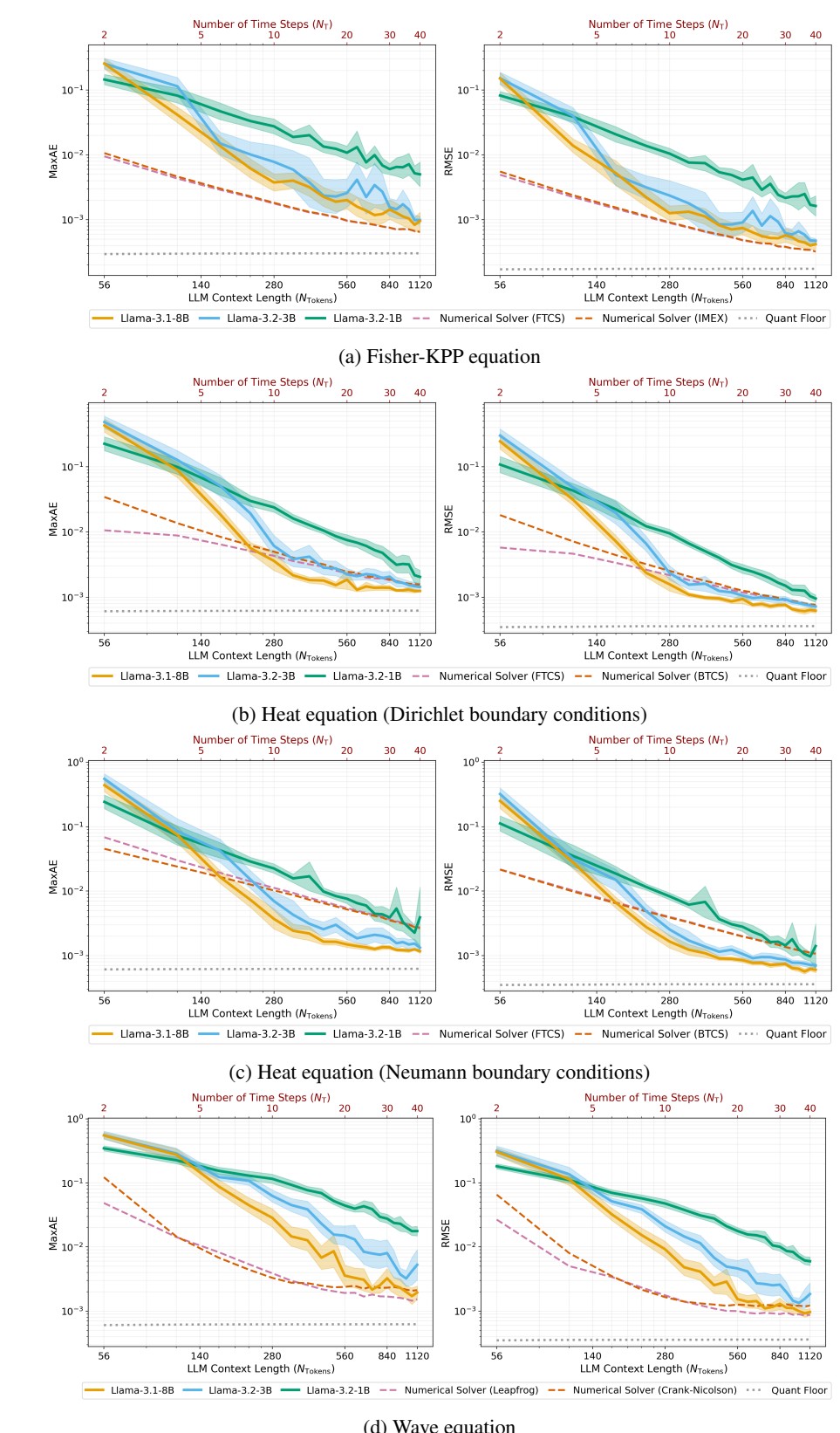

Figure 7: One-step prediction error for the Fisher–KPP, heat, and wave equations as a function of input context length, under the experimental setup of Section 4.1.

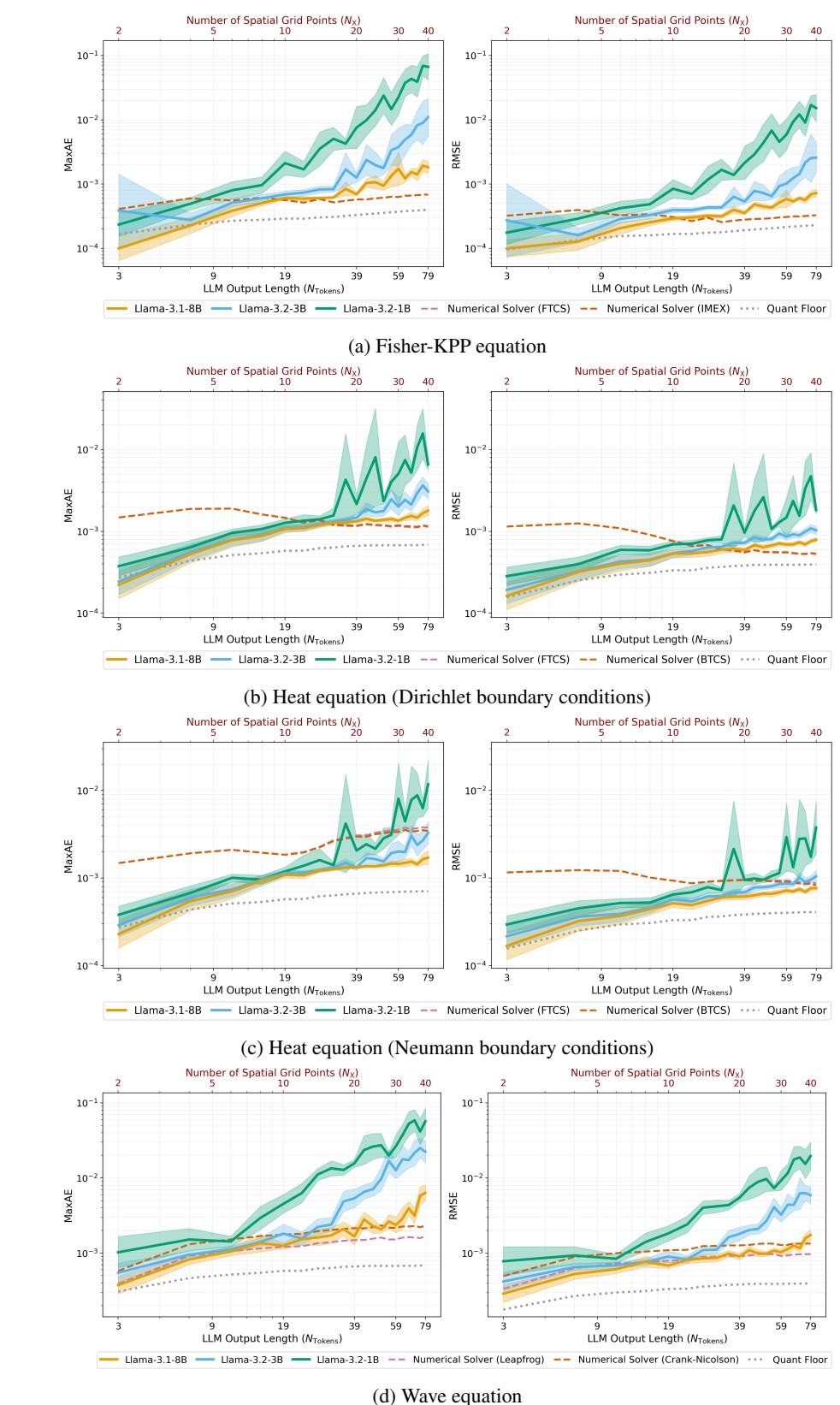

(a) Fisher-KPP equation

(b) Heat equation (Dirichlet boundary conditions)

(c) Heat equation (Neumann boundary conditions)

(d) Wave equation

Figure 8: One-step prediction error for the Fisher–KPP, heat, and wave equations as a function of output length, under the experimental setup of Section 4.1.

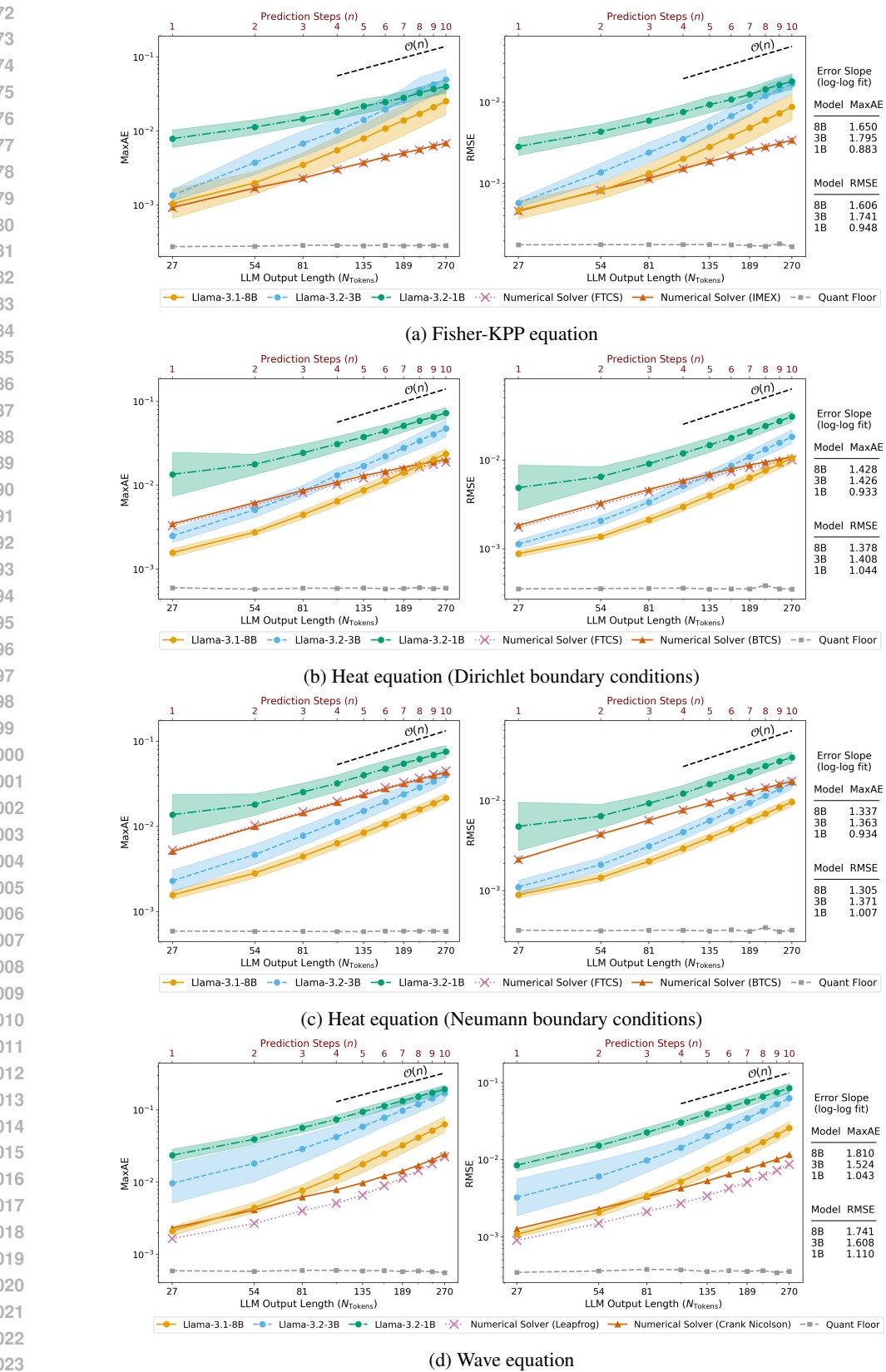

(a) Fisher-KPP equation

(b) Heat equation (Dirichlet boundary conditions)

(c) Heat equation (Neumann boundary conditions)

(d) Wave equation

Figure 9: Multi-step prediction error for the Fisher–KPP, heat, and wave equations as a function of prediction steps, under the experimental setup of Section 4.2.

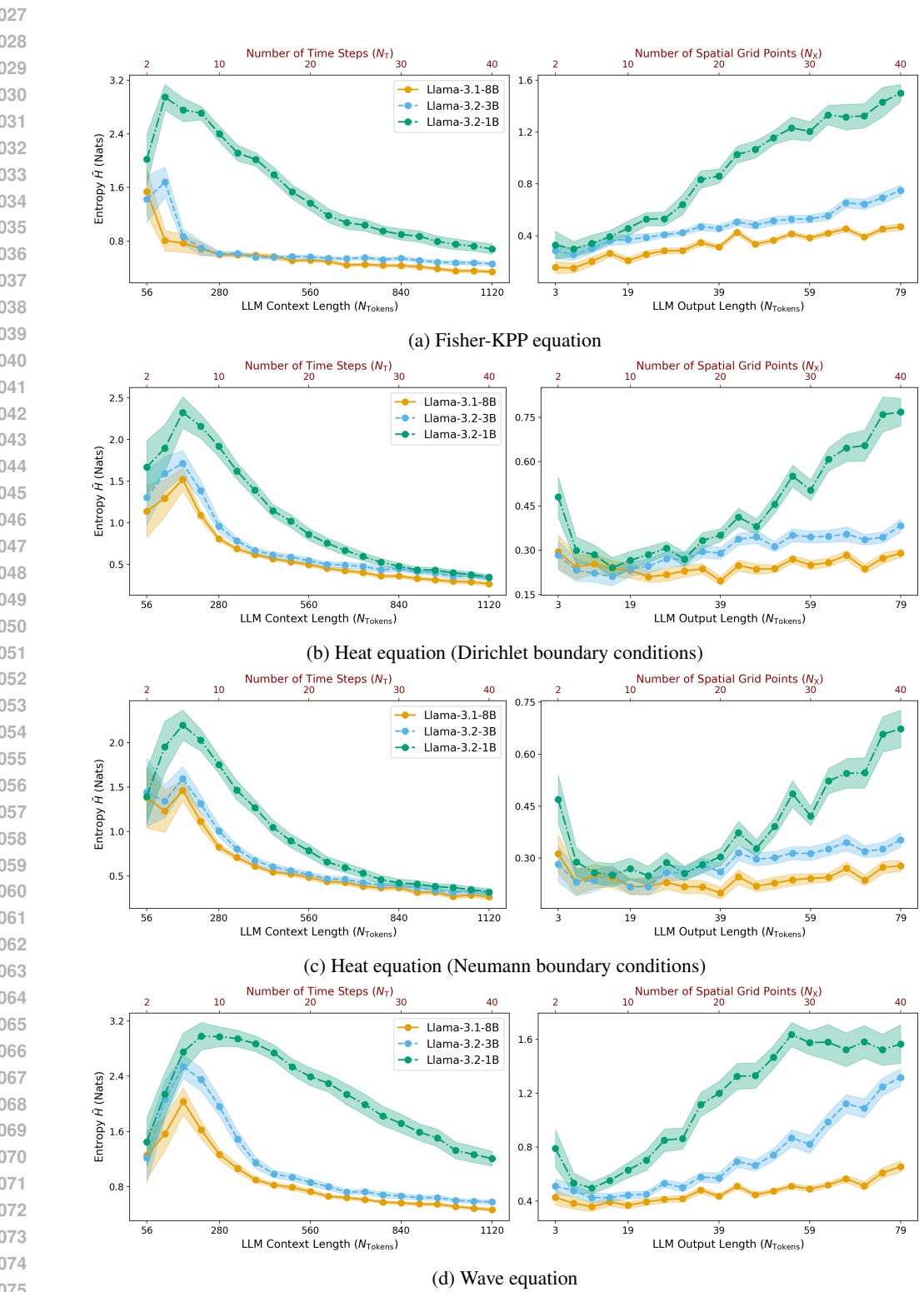

Figure 10: Uncertainty analysis of the Fisher–KPP, heat, and wave equations, under the experimental setup of Section 4.3.

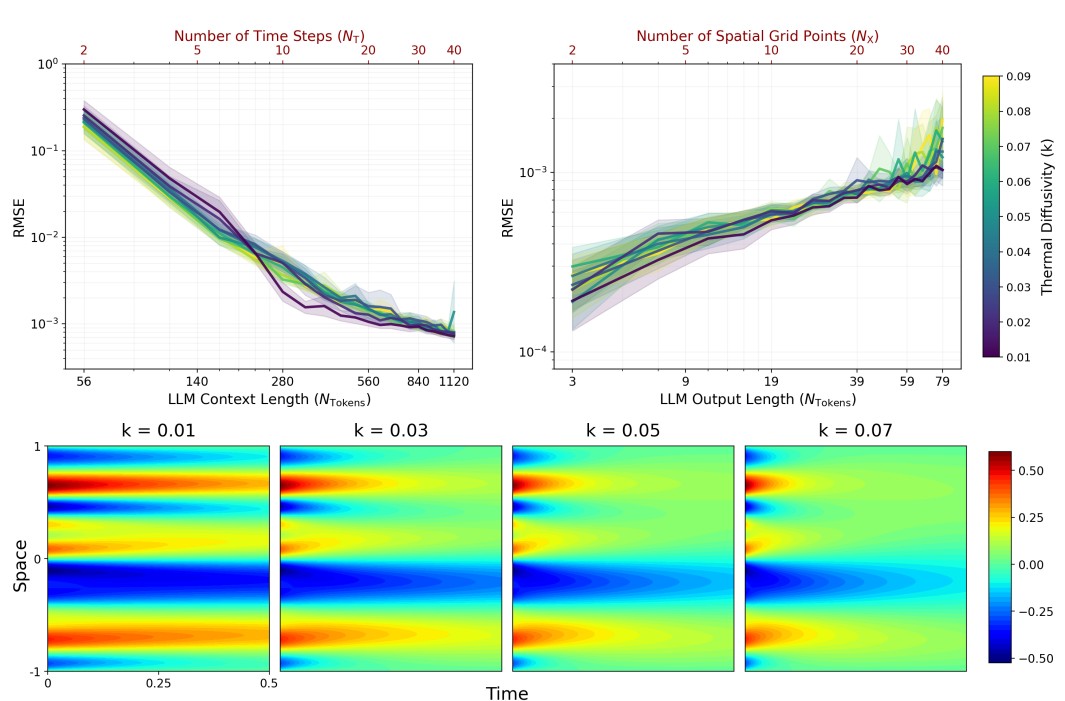

(a) Heat equation: scaling across thermal diffusivities $k$; prediction accuracy is largely insensitive to $k$. Example of the reference rollout at different $k$ for one randomly sampled initial condition.

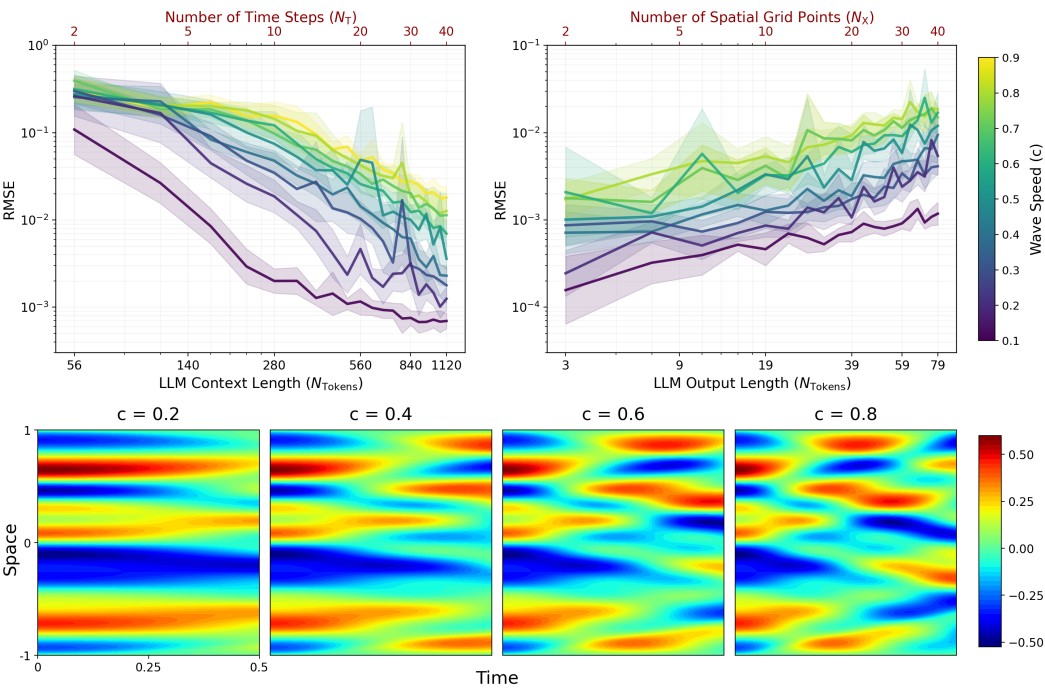

(b) Wave equation: scaling across wave speeds $c$; larger $c$ consistently degrades accuracy. Example of the reference rollout at different $c$ for one randomly sampled initial condition.

Figure 11: One-step prediction error for the Llama-3.2-3B model across varying PDE parameters, under the experimental setup of Section 4.1.

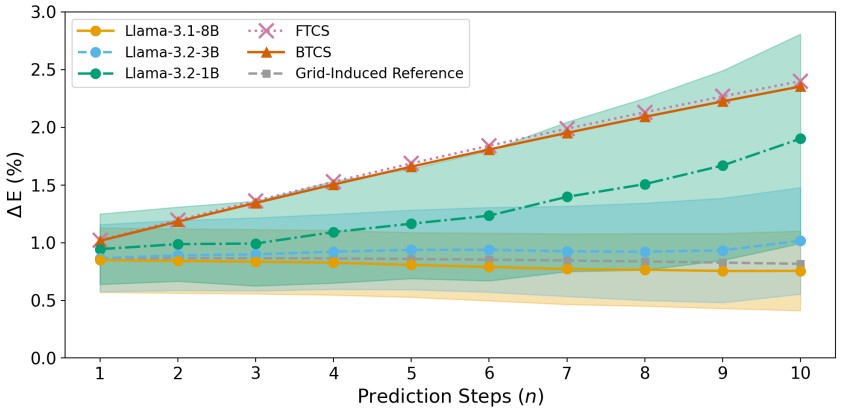

Figure 12: Relative energy deviation $\Delta E$ over prediction steps for LLM rollouts compared against classical finite difference solvers (FTCS, BTCS) under homogeneous Neumann boundary conditions for the heat equation. Shaded regions denote 95% confidence intervals over 20 random initial conditions. The grid-induced reference is computed by evaluating the total thermal energy with a high-resolution solver and then restricting it to the same coarse spatial grid used in the experimental setup; this represents the minimal deviation expected from discretization alone. Llama-3.1-8B predictions remain close to this reference and exhibit substantially lower conservation error than coarse-grid finite difference solvers across the rollout horizon.

Differentiating and using the PDE gives

$$\frac{d\mathrm{E}}{dt} = \int_{-L}^{L} \partial_t u \, dx = k \int_{-L}^{L} \partial_{xx} u \, dx = k \left[ \partial_x u \right]_{-L}^{L}.$$

The Neumann conditions enforce $\partial_x u(-L, t) = \partial_x u(L, t) = 0$, so the boundary term vanishes and hence

$$\frac{d\mathrm{E}}{dt} = 0 \quad \Rightarrow \quad \mathrm{E}(t) \equiv \mathrm{E}(0).$$

Thus, the total thermal energy is exactly conserved in the continuous dynamics.

**Relative Energy Deviation.** To evaluate conservation in rollouts, we approximate $\mathrm{E}(t)$ using the trapezoidal rule on the spatial grid $\{x_i\}_{i=0}^{N_\mathrm{X}+1}$. The grid follows the uniform setup in the main section:

$$x_i = -L + i \, \Delta x, \quad \Delta x = \frac{2L}{N_\mathrm{X} + 1}, \quad i = 0, 1, \ldots, N_\mathrm{X} + 1,$$

In our setup, since both the LLM and the finite-difference benchmarks evolve only interior values[6] $\{\hat{u}(x_i, t_j)\}_{i=1}^{N_\mathrm{X}}$, the boundary values are reconstructed by a second-order accurate approximation consistent with homogeneous Neumann conditions:

$$\hat{u}(x_0, t_j) \approx \frac{4\hat{u}(x_1, t_j) - \hat{u}(x_2, t_j)}{3}, \qquad \hat{u}(x_{N_\mathrm{X}+1}, t_j) \approx \frac{4\hat{u}(x_{N_\mathrm{X}}, t_j) - \hat{u}(x_{N_\mathrm{X}-1}, t_j)}{3}.$$

With trapezoidal weights $w_0 = w_{N_\mathrm{X}+1} = \frac{\Delta x}{2}$ and $w_i = \Delta x$ for $1 \leq i \leq N_\mathrm{X}$, the discrete energy at step $t_j$ is

$$\hat{\mathrm{E}}_j = \sum_{i=0}^{N_\mathrm{X}+1} w_i \, \hat{u}(x_i, t_j).$$

As a reference, $\mathrm{E}(0)$ is computed via high-resolution trapezoidal quadrature of the initial condition. The stepwise relative deviation is then:

$$\Delta \mathrm{E}_j = \frac{|\hat{\mathrm{E}}_j - \mathrm{E}(0)|}{|\mathrm{E}(0)|} \times 100\%.$$

---

[6]This convention is standard in finite-difference schemes, where boundary values are imposed rather than evolved. In our setup, this makes the task more challenging for the LLM: for example, in Dirichlet problems, boundary values are not given explicitly but must be inferred from the interior evolution.

Here $\Delta \mathrm{E}_j = 0$ corresponds to exact conservation, while nonzero values measure violations at prediction step $t_j$. This metric parallels $\mathrm{RMSE}_j$ and $\mathrm{MaxAE}_j$, enabling direct comparison between accuracy and conservation fidelity.

Figure 12 reports results under the same setup as the Neumann-boundary heat equation experiments shown in Figure 9. The only modification is that initial conditions are drawn from $a = 0, b = 1$ (instead of $a = -0.5, b = 0.5$), following Appendix A.1.2, so that the conserved energy $\mathrm{E}(0)$ is bounded away from zero. This ensures numerical stability when evaluating relative deviations. We also verified that one-step, multi-step, and uncertainty-evolution analyses under Neumann boundaries with $a = 0, b = 1$ show consistent qualitative behavior with the results in Appendix A.3.1 (which use $a = -0.5, b = 0.5$); to avoid redundancy, we do not reproduce these plots here, but the full results are available in the accompanying GitHub repository. Under this setup, LLM rollouts maintain thermal energy close to its conserved value across time, and notably more faithfully than coarse finite difference solvers operating at the same resolution.

This conservation behavior reflects more than numerical accuracy. When continuing spatiotemporal PDE trajectories, the model is not merely extrapolating forward in time or interpolating across space in isolation; rather, it is simultaneously inferring both spatial structure and temporal evolution from in-context information. The fact that thermal energy remains close to its conserved value under Neumann boundaries indicates that ICL can internalize and propagate governing conservation principles, rather than relying on surface-level pattern imitation or naive extrapolation.

## A.4 RESULTS FOR INSTRUCTION-TUNED LLAMA-3 VARIANTS

We replicate the full experimental setup from Section 4 using instruction-tuned variants of Llama-3 models: Llama-3.1-8B-Instruct, Llama-3.2-3B-Instruct, and Llama-3.2-1B-Instruct. These models are instruction-tuned for assistant-like chat, whereas the pretrained base models are designed to support a broader range of natural language generation tasks (Grattafiori et al., 2024). All evaluations (one-step prediction, multi-step rollout, and entropy-based uncertainty quantification) are conducted using the same setup as described in Section 4.

As shown in Figure 13, the qualitative trends closely mirror those of the base models. In particular, in one-step prediction settings, accuracy improves with longer input context and degrades with increasing output length. In multi-step prediction settings, predictions exhibit algebraic error accumulation. Similarly, Figure 14 shows that prediction uncertainty undergoes stage-wise transitions as input context increases and grows with output length. Notably, under the same accuracy evaluation setup, the smaller instruction-tuned models, Llama-3.2-3B-Instruct and Llama-3.2-1B-Instruct, show reduced prediction accuracy compared to their pretrained base counterparts. This observation is consistent with prior findings from Gruver et al. (2023), which suggest that alignment procedures such as instruction tuning and Reinforcement Learning with Human Feedback (RLHF) can adversely affect time-series forecasting performance in Llama-2 models. In contrast, we do not observe such a negative impact on the 8B instruction-tuned variant, indicating that newer, larger models may be more robust to the effects of alignment in the context of continuing the spatiotemporal dynamics of PDEs.

## A.5 RESULTS FOR OTHER MODEL FAMILIES

We replicate the full experimental setup from Section 4 using models outside the Llama family: Phi-4-14B and SmolLM-3-3B, with Llama-3.2-3B (the representative model analyzed in the main text) included for comparison. Figures 15 and 16 summarize the results.

As shown in Figure 15, the overall qualitative trends remain consistent with those reported in the main text. One-step prediction accuracy improves with longer input context and degrades with increasing output length, while multi-step rollouts exhibit algebraic error accumulation. For models with similar parameter counts—e.g., Llama-3.2-3B and SmolLM-3-3B—the exact quantitative prediction errors differ slightly. This reinforces the observation from the main text that the "model size effect" we report arises primarily when comparing models within the same family; models from different families with similar parameter sizes can exhibit slightly different prediction errors, likely reflecting differences in architecture, training data, and other design choices.

Similarly, Figure 16 shows that the prediction uncertainty trends are consistent with those observed for Llama-3 models. Specifically, spatial entropy progresses through three distinct learning stages as

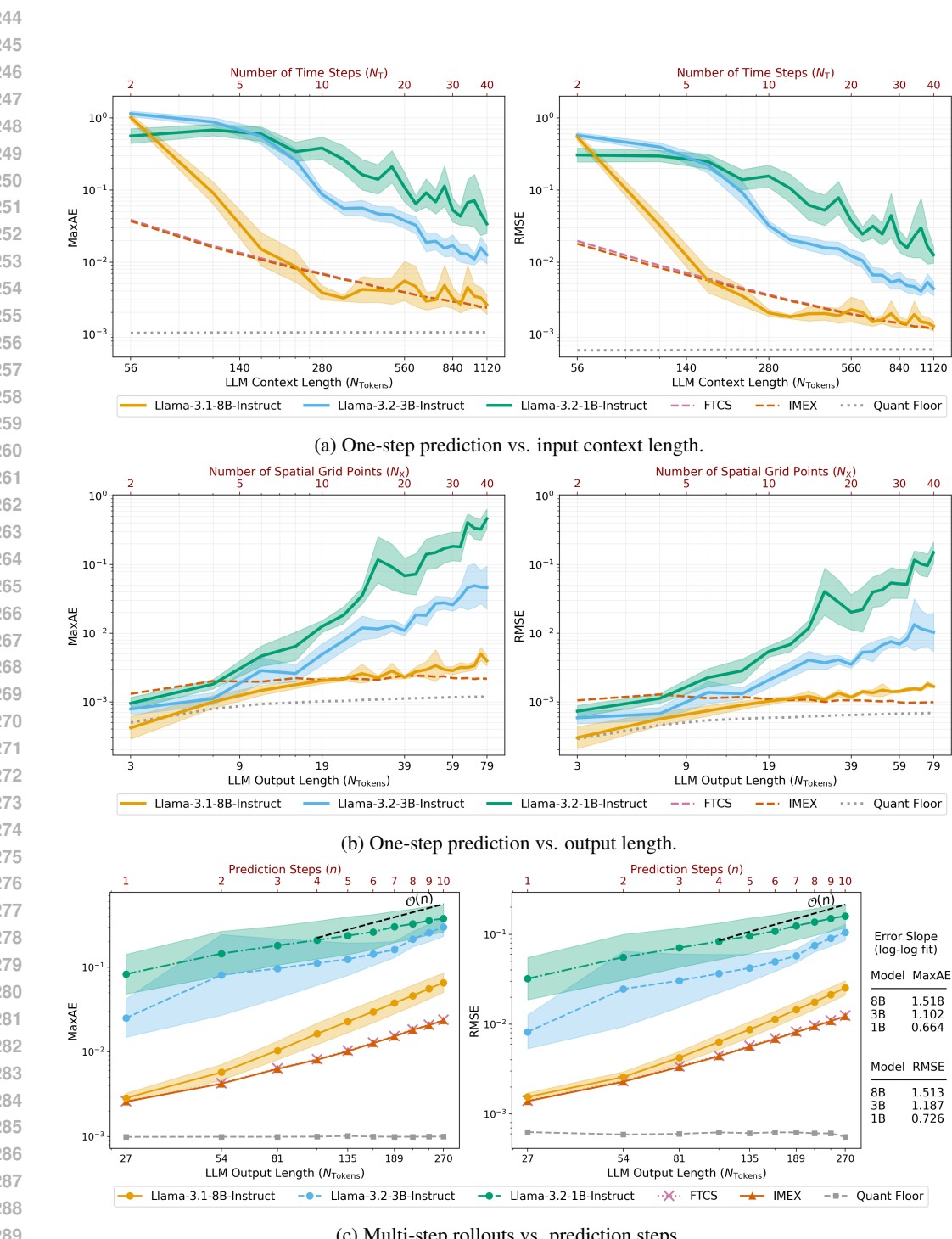

(a) One-step prediction vs. input context length.

(b) One-step prediction vs. output length.

(c) Multi-step rollouts vs. prediction steps.

Figure 13: Prediction accuracy of instruction-tuned Llama-3 models, using the same experimental setup described in Sections 4.1 and 4.2.

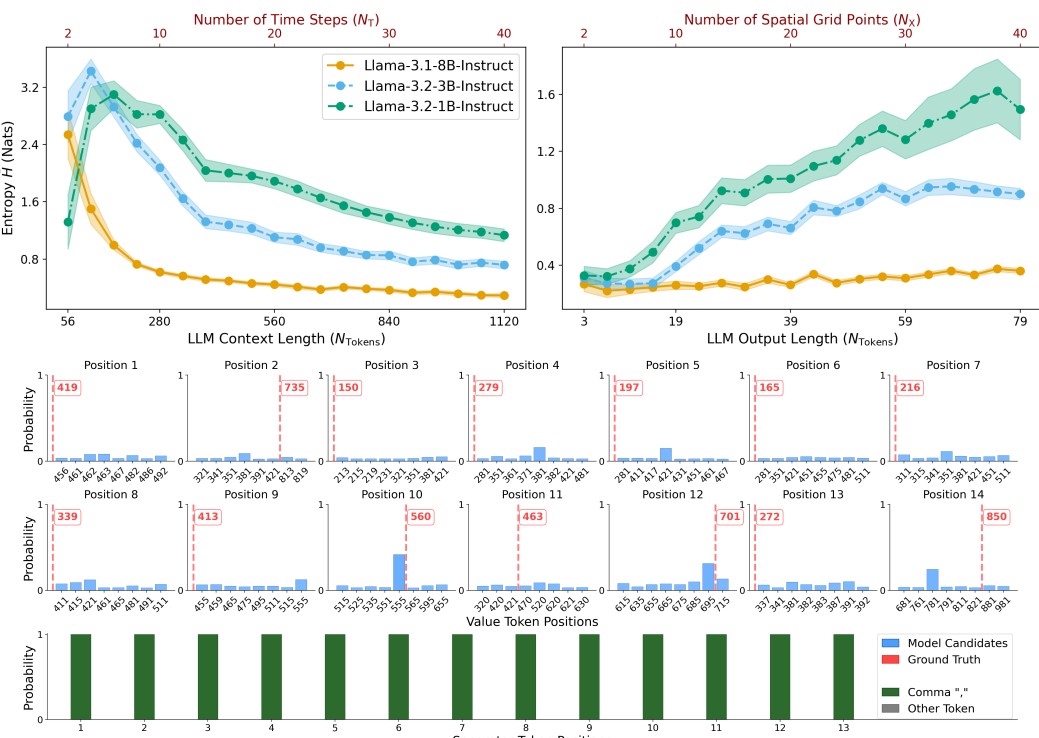

Figure 14: Uncertainty analysis of instruction-tuned Llama-3 models, using the same experimental setup described in Section 4.3 and Figure 5. **Top**: Mean spatial entropy $\bar{H}$ as a function of context length $N_{\mathrm{T}}$ (left) and output length $N_{\mathrm{X}}$ (right). **Bottom**: Token-level softmax distributions at the syntax-only stage for Llama-3.1-8B-Instruct. The overall entropy behavior remains consistent with the pretrained base model, with one notable difference: at very short contexts ($N_{\mathrm{T}} = 2$), the instruction-tuned 8B variant exhibits higher average spatial entropy, reflecting greater uncertainty under minimal context. However, separator tokens (e.g., commas) are still predicted with near-perfect confidence, and this increased uncertainty arises from spatial value tokens more frequently acting as generic placeholders rather than producing deterministically incorrect outputs, as observed in the base model.

input context increases, grows with output length, and separator tokens (e.g., commas) are predicted with near-perfect confidence across all settings. The main deviation is that Phi-4-14B consistently exhibits higher mean spatial entropy than the two 3B models. This difference is largely attributable to the inference temperature: Llama-3[7] and SmolLM-3[8] use $T = 0.6$, the default generation temperature specified in their configuration files, while Phi-4[9] defaults to $T = 1.0$, as its configuration omits a temperature setting. Lower temperatures ($T < 1$) scale the logits to increase their relative magnitude, yielding sharper (lower-entropy) output distributions. In contrast, higher temperatures ($T > 1$) reduce these relative differences, producing flatter (higher-entropy) distributions. The temperature-scaled softmax function (Goodfellow et al., 2016) is defined as:

$$\mathrm{softmax}(\boldsymbol{z}; T)_i = \frac{\exp(z_i/T)}{\sum_j \exp(z_j/T)},$$

where $z_i$ denotes the $i$-th logit prior to normalization.

---

[7]https://huggingface.co/meta-llama/Llama-3.2-3B/tree/main
[8]https://huggingface.co/HuggingFaceTB/SmolLM3-3B/tree/main
[9]https://huggingface.co/microsoft/phi-4/tree/main

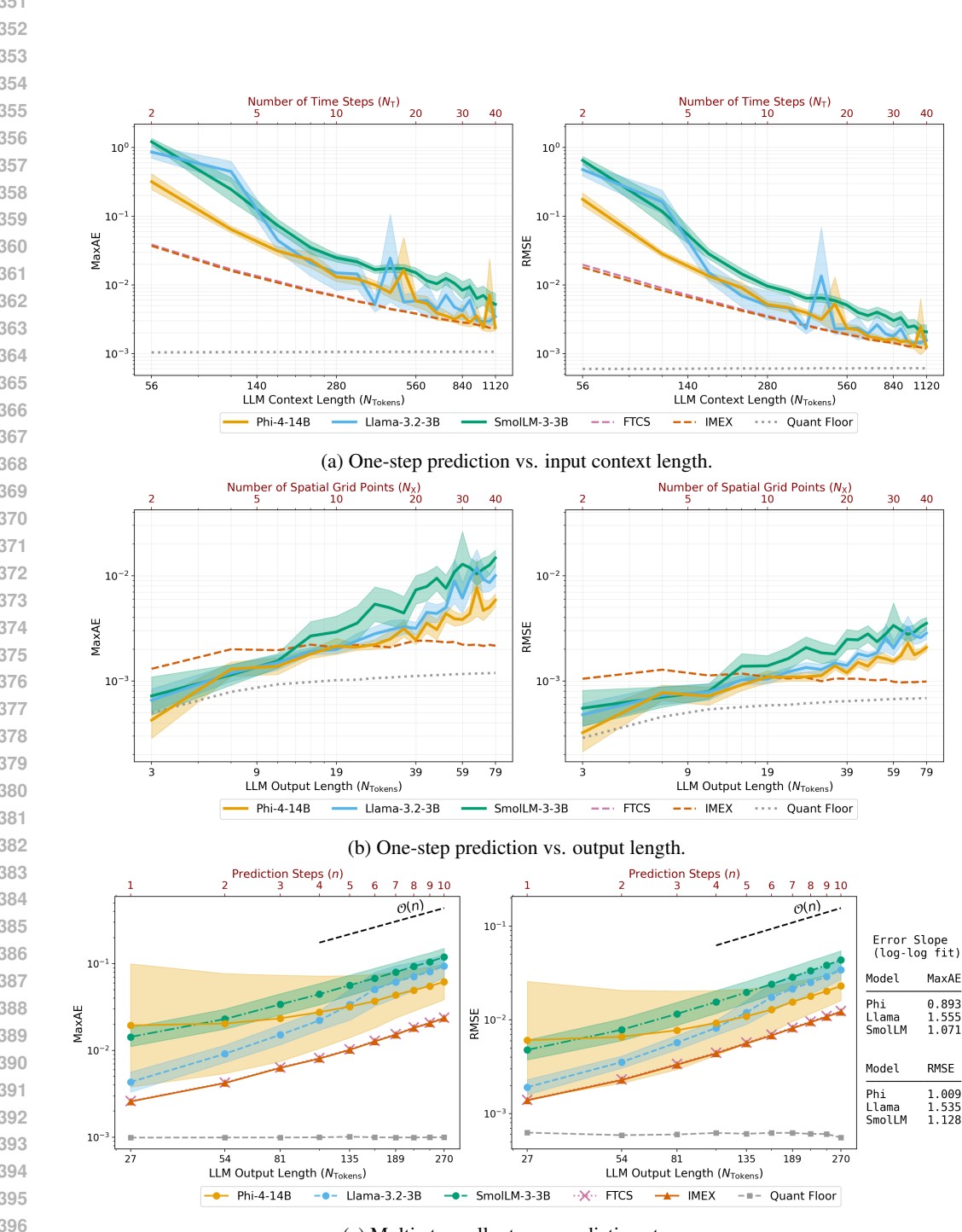

(a) One-step prediction vs. input context length.

(b) One-step prediction vs. output length.

(c) Multi-step rollouts vs. prediction steps.

Figure 15: Prediction accuracy of Phi-4-14B and SmolLM-3-3B models (with Llama-3.2-3B for reference), using the same experimental setup described in Sections 4.1 and 4.2.

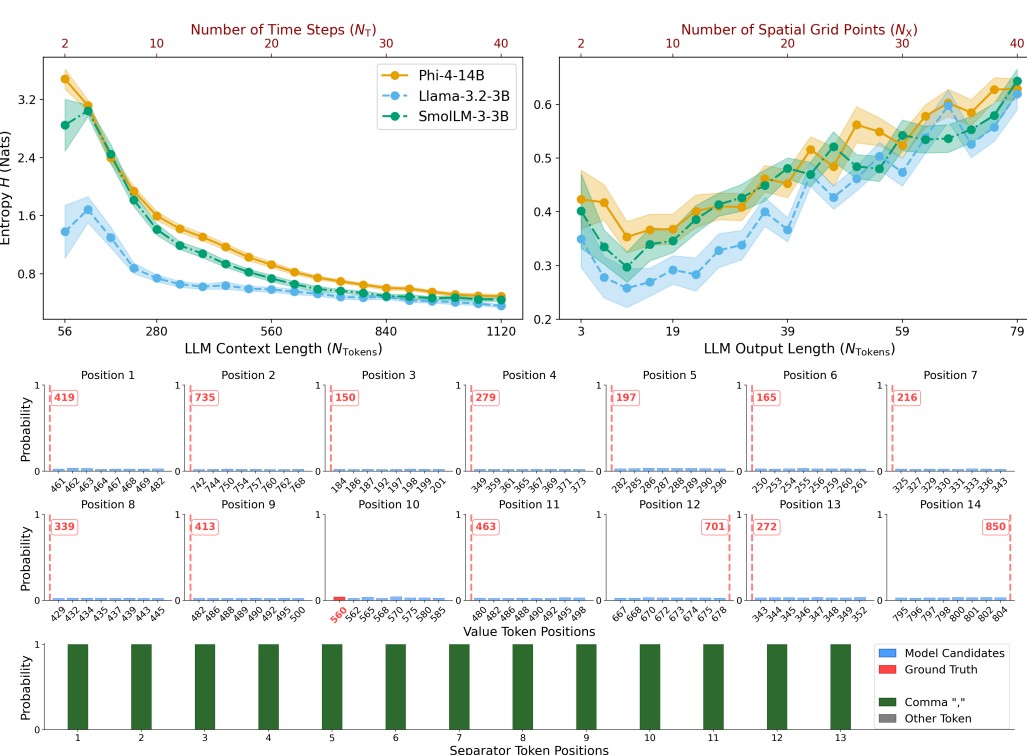

Figure 16: Uncertainty analysis of Phi-4-14B and SmolLM-3-3B models, with Llama-3.2-3B included for reference, using the same experimental setup described in Section 4.3 and Figure 5. **Top**: Mean spatial entropy $\bar{H}$ as a function of context length $N_T$ (left) and output length $N_X$ (right). **Bottom**: Token-level softmax distributions at the syntax-only stage for Phi-4-14B. The overall entropy behavior remains consistent with the Llama-3 models, with one notable difference: Phi-4-14B exhibits higher average spatial entropy than the two 3B models shown here, in nearly all cases, due to its higher default inference temperature (see Appendix A.5 for discussion). A pronounced peak in average spatial entropy also appears at very short contexts ($N_T = 2$), similar to Llama-3.1-8B-Instruct, where increased uncertainty arises from spatial value tokens more frequently acting as generic placeholders rather than producing deterministically incorrect outputs.

Table 1: Architectural details of evaluated Llama-3 models

| Parameter | Llama-3.1-8B | Llama-3.2-3B | Llama-3.2-1B |
|---|---|---|---|
| Hidden size | 4096 | 3072 | 2048 |
| Hidden layers | 32 | 28 | 16 |
| Head dimension | 128 | 128 | 64 |
| Attention heads | 32 | 24 | 32 |
| Intermediate size | 14336 | 8192 | 8192 |
| Tie word embeddings | false | true | true |
| RoPE scaling factor | 8.0 | 32.0 | 32.0 |
| Transformers version | 4.43.0.dev0 | 4.45.0.dev0 | 4.45.0.dev0 |
| *Shared configurations* | | | |
| Architecture | | LlamaForCausalLM | |
| Attention bias | | false | |
| Attention dropout | | 0.0 | |
| BOS token ID | | 128000 | |
| EOS token ID | | 128001 | |
| Activation function | | SiLU | |
| Initializer range | | 0.02 | |
| Max position embeddings | | 131072 | |
| MLP bias | | false | |
| Model type | | llama | |
| Key-value heads | | 8 | |
| Pretraining tp | | 1 | |
| RMS norm $\epsilon$ | | $10^{-5}$ | |
| RoPE scaling low frequency factor | | 1.0 | |
| RoPE scaling high frequency factor | | 4.0 | |
| RoPE scaling original max position embeddings | | 8192 | |
| RoPE scaling type | | llama3 | |
| RoPE theta | | 500000.0 | |
| Torch dtype | | bfloat16 | |
| Use cache | | true | |
| Vocabulary size | | 128256 | |

## A.6 ARCHITECTURAL DETAILS AND SIZE COMPARISON OF LLAMA-3 MODELS

Table 1 summarizes the architectural configurations of the three Llama-3 models evaluated in our main experiments. All values are sourced directly from Meta's official `generation_config` files, released on Hugging Face.[10] Differences in parameters such as hidden size, number of layers, and attention configurations account for the varying model sizes, which in turn influence the emergence of ICL behaviors observed in zero-shot solutions of PDEs.

## A.7 VALIDATION OF NON-TRIVIAL TEMPORAL EVOLUTION

To empirically validate that our prediction tasks (Section 3) are non-trivial, we analyze the temporal differences $Q_{i,j+1} - Q_{i,j}$ across the spatial grid under the finest discretization setting ($N_X = 40$, $N_T = 50$). As shown in Figure 17, the discretized solution exhibits meaningful variation between adjacent time steps across the spatial domain, confirming that the system evolves in a non-trivial manner over time. Coarser discretizations (e.g., $N_X = 14$, $N_T = 25$), as used in the multi-step rollout task, naturally introduce larger changes between time steps due to increased temporal spacing compared to the finest setting shown here. These observations support the discretization design choices used in the main experiments, which preserve clear spatiotemporal variation and ensure that model performance reflects an understanding of the underlying PDE dynamics rather than relying on trivial extrapolation strategies.

---

[10]https://huggingface.co/meta-llama

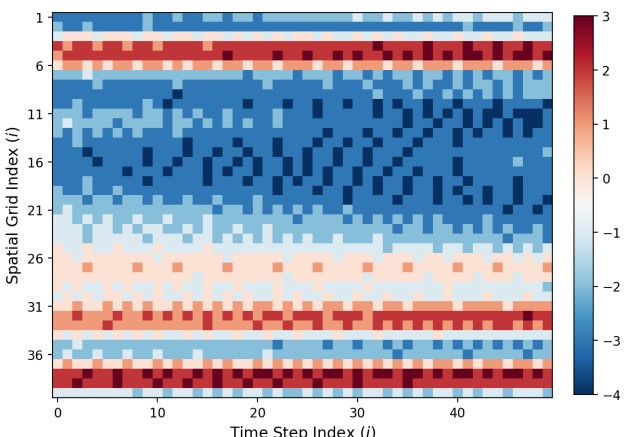

Figure 17: Temporal differences $Q_{i,j+1} - Q_{i,j}$ at each spatial grid point for the finest discretization setting ($N_X = 40$, $N_T = 50$) used in the experimental setups of Section 4, where $\boldsymbol{Q} \in \mathcal{Z}^{N_X \times (N_T+1)}$ is the quantized representation of the PDE solution used as input to the LLM (see Section 3). The heatmap shows that, even at this resolution, the discretized solution exhibits meaningful changes between adjacent time steps, indicating that the prediction task requires modeling nontrivial temporal evolution. Since coarser discretizations correspond to larger time steps, they naturally induce greater local variation, further supporting the non-trivial nature of the prediction task across discretizations.

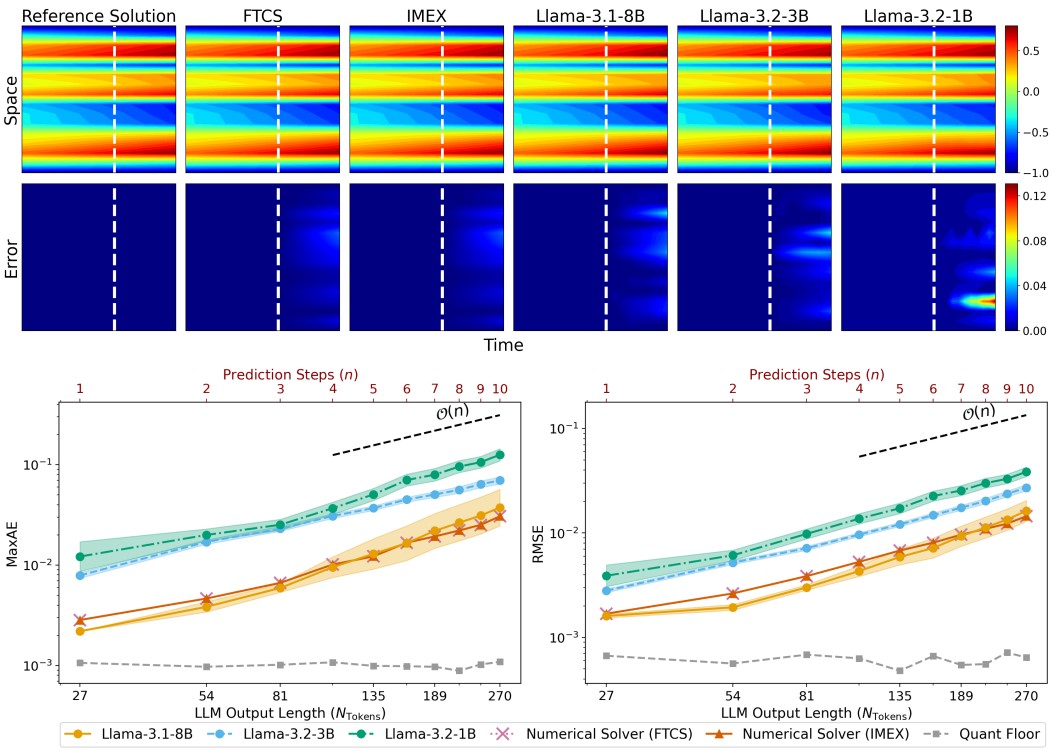

Figure 18: Multi-step prediction and error visualization for the Allen–Cahn equation with a different randomly sampled initial condition, using the same experimental setup as in Section 4.2.

## A.8 ADDITIONAL MULTI-STEP ROLLOUT VISUALIZATIONS

Figure 18 presents the multi-step prediction error trend for the Allen–Cahn equation with a different initial condition, sampled with `np.random.seed(42)` following the procedure in

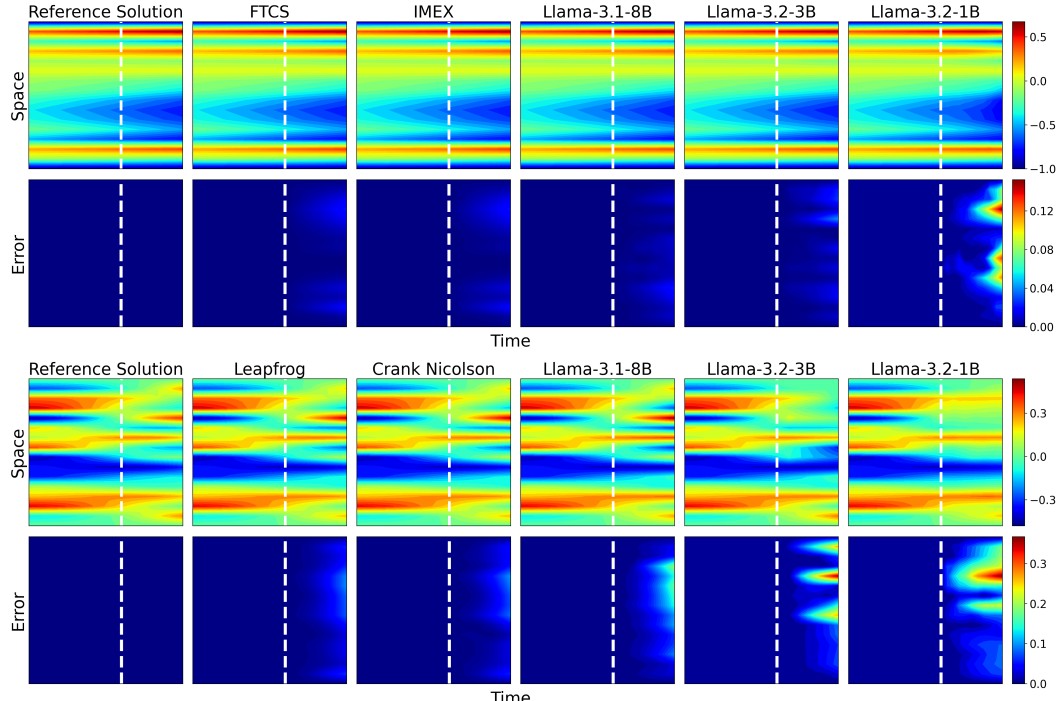

Figure 19: Detailed visualizations corresponding to the main-text example in Figure 3, with additional numerical benchmarks and results from the Llama-3.2-3B model.

Appendix A.1. The main text (Figure 3) uses `np.random.seed(1)` for Allen–Cahn and `np.random.seed(42)` for wave. Changing the seed alters the sampled interior values, yielding distinct spatiotemporal trajectories. The results here corroborate those in Section 4.2, indicating that the observed model behaviors generalize across initial conditions.

Figure 19 provides the detailed visualizations corresponding to the main-text example in Figure 3. It shows representative multi-step rollouts for the Allen–Cahn and wave equations with one additional numerical benchmark each: IMEX for Allen–Cahn and Crank–Nicolson for the wave equation. This figure also includes results for the Llama-3.2-3B model, which were omitted from the main text due to space constraints. The extended visualizations further illustrate that the smallest model (Llama-3.2-1B) struggles to sustain coherent PDE dynamics over extended horizons, while numerical benchmarks confirm consistency with classical solvers.

## A.9 ERROR PATTERNS AND CAPACITY LIMITATIONS IN THE LLAMA-3.2-1B MODEL

In this appendix, we analyze systematic errors exhibited by the smallest model, Llama-3.2-1B, during multi-step PDE rollouts. Figure 20 shows predictions averaged over 20 LLM repeats for the randomly sampled initial condition in the main text alongside bias analyses of the resulting errors. The 1B model produces structured errors that grow over the 10-step prediction horizon and consistently concentrate at specific spatial locations, in contrast to the low-magnitude errors from the 3B and 8B models, which remain more evenly distributed across the spatial domain. These persistent error patterns from the 1B model, averaged over 20 repeats from the same initial condition, suggest a systematic prediction bias likely attributable to the model's inductive biases or capacity limitations, rather than to stochasticity introduced during LLM sampling at inference time.

The bottom panels of Figure 20 investigate the origin of this systematic prediction bias by analyzing how the prediction errors relate to properties of the discretized PDE solution—specifically, its magnitude and local variation in space and time. The left panel shows errors grouped by the reconstructed floating-point values of the 1B model's outputs. The lack of any discernible trend suggests that the bias is not driven by output magnitude or tokenization artifacts such as a preference for

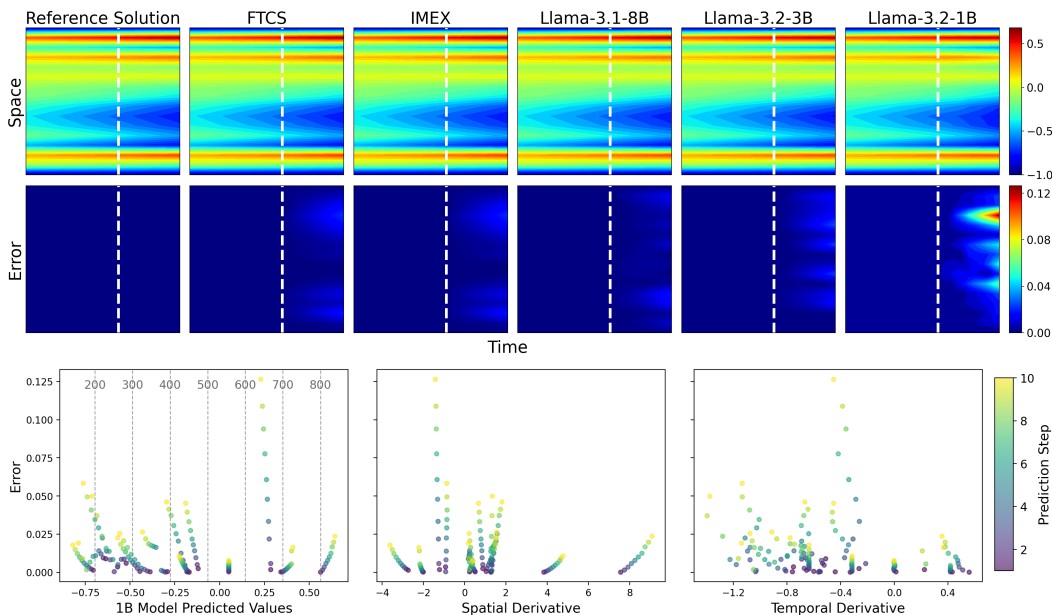

Figure 20: Multi-step rollouts and error analysis for the Llama-3.2-1B model. **Top**: Predictions averaged over 20 LLM repeats for the randomly sampled initial conditions used in the main text. **Bottom**: Bias analysis. Left: Absolute error vs. reconstructed prediction value shows no correlation, indicating that errors are not biased toward particular magnitudes or integer-like values (e.g., 200, 300, 400), suggesting tokenization procedures are unlikely to be the source. Center and right: Errors vs. spatial and temporal derivatives. In this rollout, larger errors cluster in regions of low variation, though this trend does not persist across other initial conditions.

special integer-like values (e.g., 200, 300, 400). The center and right panels display errors grouped by spatial and temporal derivatives, respectively, both approximated using finite-difference stencils (central differences for interior points and forward/backward differences at boundaries). In this specific rollout, higher errors tend to occur in regions with lower local variation, hinting at a potential trend. However, this behavior is not consistent across different initial conditions.

These findings help rule out tokenization effects as well as simple correlations with solution magnitude or local variation as the primary sources of bias, but do not conclusively identify its origin. Whether the bias arises from limited model capacity, inductive priors, or specific dynamical regimes that are inherently more difficult for smaller models to internalize remains an open question for future investigation.

## A.10 ACCUMULATION OF PREDICTIVE UNCERTAINTY IN MULTI-STEP ROLLOUTS

In this appendix, we extend the entropy-based uncertainty analysis described in Section 4.3 for one-step predictions to the multi-step rollout setup introduced in Section 4.2, where the LLM autoregressively generates future time steps by appending its own outputs as additional inputs at each step. Since no natural language prompting is used, the model does not distinguish between ground-truth context and its own generated predictions, allowing us to directly analyze its ICL capacity to roll out PDE dynamics purely from serialized numerical input.

Under fixed spatial and temporal discretization, all models exhibit a consistent decrease in mean spatial entropy $\bar{H}$ with increasing prediction horizon (Figure 21), reflecting progressively more deterministic outputs. Notably, the smaller model (Llama-3.2-1B) maintains substantially higher entropy throughout the rollout compared to larger models (Llama-3.1-8B and Llama-3.2-3B), indicating that the model size influences the confidence level of multi-step predictions. However, this trend toward increased confidence does not correspond to improved predictive accuracy: as shown in Figure 4, errors grow algebraically over time. This illustrates that prediction uncertainty may decrease due to internal belief reinforcement within the LLM, even as predictive accuracy degrades.

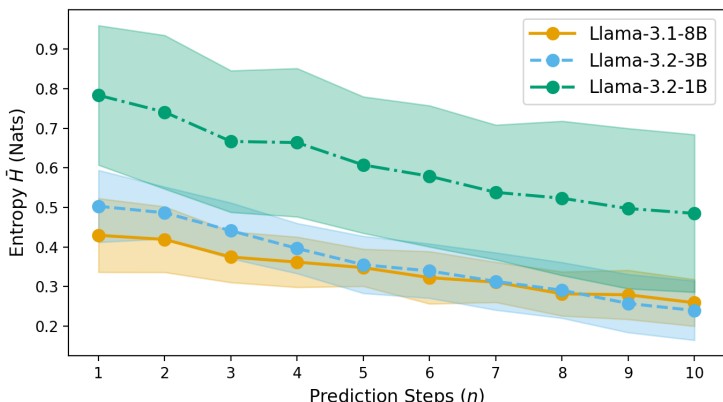

Figure 21: Mean spatial entropy $\bar{H}$ as a function of prediction step $n$, using the multi-step rollout setup from Section 4.2 and the uncertainty metric defined in Section 4.3. Shaded regions denote 95% confidence intervals across 50 randomly sampled initial conditions, with one LLM rollout for each initial condition.

## A.11 Token-Level Distribution Visualizations Across Learning Stages

To complement the stage-wise analysis in Section 4.3, Figure 22 visualizes token-level softmax distributions from Llama-3.1-8B across all three ICL stages—syntax-only, exploratory, and consolidation—for a representative initial condition (same as in the main text). The syntax-only stage is shown in the bottom panel of Figure 5; this appendix provides a more comprehensive view by including the later stages as well. Together, these illustrate how the model's predictive uncertainty evolves with increasing context length: initially focused on reproducing surface-level syntax, then entering a phase of high uncertainty as it explores plausible continuations, and ultimately converging to confident predictions that align with the underlying spatiotemporal PDE dynamics.

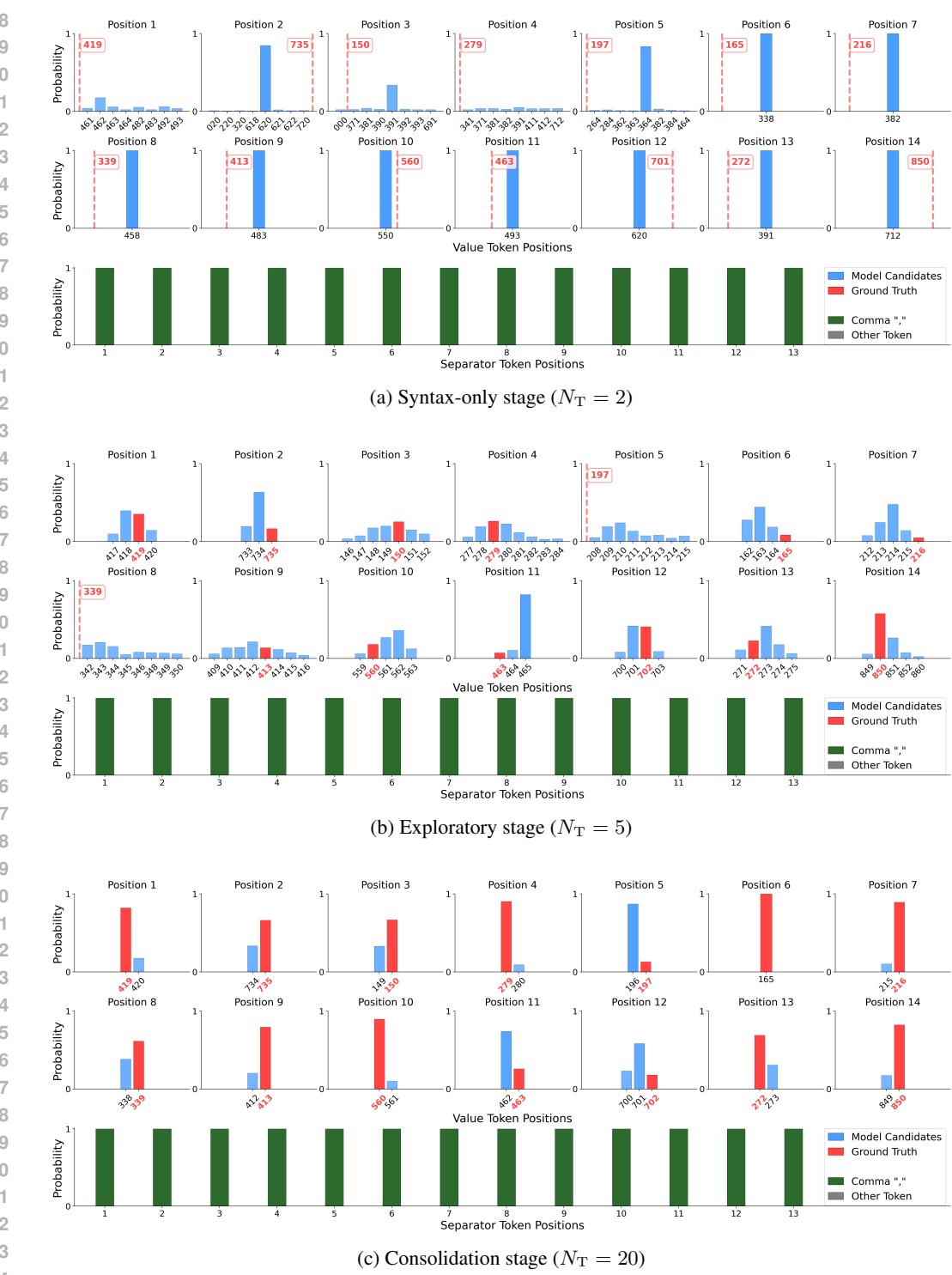

Figure 22: Representative token distributions across ICL stages, extracted from the Llama-3.1-8B model's softmax outputs on a randomly sampled initial condition (same as in the multi-step rollout example). For clarity, only the top 8 candidate tokens (by probability) are shown per spatial position. (a) Syntax-only stage: separator tokens (e.g., commas) are predicted with near-perfect confidence, while spatial values are either deterministic but incorrect or act as generic placeholders. (b) Exploratory stage: spatial value distributions broaden, reflecting increased uncertainty and competing hypotheses with partial alignment to ground truth. (c) Consolidation stage: uncertainty decreases, and distributions sharpen around the true target values.