# OpenReview forum: "Text-Trained LLMs Can Zero-Shot Extrapolate PDE Dynamics"
_ICLR.cc/2026/Conference — ICLR 2026 Conference Withdrawn Submission_

### Official Review · Reviewer_Z8eQ · 2025-10-20

**Soundness:** 3
**Presentation:** 3
**Contribution:** 3
**Rating:** 6
**Confidence:** 3

**Summary:**

The paper studies the ability of pre-trained LLMs to extrapolate PDE solutions given the solution trajectory up to some time. The paper investigates how the error and the uncertainty in the model’s prediction varies as the context length, rollout length, and spatial discretization are all varied. In the one-step prediction setting, they find that the mean error decay exhibits different behaviors depending on the context length, and that for sufficiently long contexts, the error decays similarly to a numerical solver. They also find that, unlike numerical solvers, the LLM’s prediction error increases as the spatial resolution is increased, since this results in a larger token length. In the multi-step prediction setting, they find that errors accumulate algebraically along the predicted trajectory, which is also observed in numerical solvers. Finally, they find that the uncertainty of the model’s prediction (as measured by the Shannon entropy) decreases as the context length increases and increases as the token length increases.

**Strengths:**

- The paper provides a very thorough empirical study of the scaling laws for solving PDEs with pre-trained LLMs, and the fact that you see similar behavior to classical numerical solvers across various initial conditions makes me feel better about the use of LLMs as PDE solvers.
- Uncertainty quantification for LLMs is a very important problem, especially in scientific domains where interpretability is valued. I find it very interesting and encouraging that increasing context length reduces the uncertainty of the model’s prediction in this setting.
- The paper is very well-written, and it is easy to interpret their figures and understand their central claims.

**Weaknesses:**

- The collection of PDEs considered is somewhat limited, since the parameters of the nonlinear PDEs are fixed.
- There is no theory to explain the empirical results, though of course a theory of modern LLMs is beyond our current scope.

**Questions:**

- For one-step prediction, does the RMSE take into account the scale or the smoothness of the ground truth solution? It seems like it does not. I don’t think the choice of metric is too important, since ultimately you are comparing the model’s error with the numerical solver. But it occurs to me that, if your solution is very smooth in time, the model could just output the final token of the context and achieve $O(\Delta t)$ error (maybe this is what the model does at intermediate context lengths?).
- I would be interested to see how the LLM’s prediction depends on the choice of discretization, e.g., what happens if you instead represent the solution as a Fourier series (in $x$) and truncate the number of terms.
- The RMSE and MAE approximate the $L^2$ and $\sup$-norms on the function space. Do you have any sense of how well the model is predicting the (spatial) derivative of the solution at each time (i.e., what the $C^1$ or $H^1$ looks like)? I guess this is related to my previous question - since the model output does not define a continuous function, it is difficult to take derivatives, but you could do this if you had the Fourier coefficients.

---

### Official Review · Reviewer_CtL5 · 2025-10-27

**Soundness:** 2
**Presentation:** 2
**Contribution:** 1
**Rating:** 2
**Confidence:** 3

**Summary:**

The authors propose a novel and ambitious approach to solving Partial Differential Equations (PDEs) by using text-trained Large Language Models (LLMs) in a zero-shot setting. The central claim is that these models, despite being trained purely on natural language, can "extrapolate" to solve unseen PDE problems, ostensibly by reasoning over the textual representation of the equations. The paper investigates this capability, through multiple empirical experiments. They show that 1) LLM's are able to correctly perform next-step and multi-step prediction of time-dependent PDEs, when a large context window is given. 2) They show that uncertainty decreases with the LLM context and 3) uncertainty growth with output length.

The main contributions are:

- they demonstrate LLMs exhibit zero-shot predictive capabilities.
- they in-context scaling laws for time-dependent PDEs with respect to context length and spatial discretization, ...
- they analyze token-level predictive entropy.

**Strengths:**

I think the idea of using LLMs for time-dependent PDEs is an interesting area of research and is worth investigating. This is intriguing question: can the reasoning capabilities of LLMs, developed through language modeling, be transferred to a complex, formal domain like PDEs without any domain-specific fine-tuning?

Globally, the paper is well written and well presented.

**Weaknesses:**

Here is a more direct rephrasing of your points:

The paper's contributions and experiments are relatively weak. The finding that error decreases with context length is an expected outcome for next-step prediction in LLMs; a larger context simply provides better statistics for pattern matching. This result offers little evidence of a genuine ability to extrapolate PDE dynamics. Furthermore, the multi-step prediction results (e.g., Figure 4) show error increasing rapidly, which undermines the extrapolation claim by demonstrating the model fails to capture the underlying governing principles.

The core premise seems based on a misunderstanding of what LLMs learn. These models are trained on token co-occurrence, not to derive or understand first-principles physics. There is no strong theoretical justification for why a model trained on web text should be able to solve PDEs. This is an important question that should be treated to justify LLMs can perform zero-shot on PDE dynamics.

Finally, the method has only been tested on simple PDEs over a few time steps. Given these limited results, the title and its "zero-shot extrapolation" claim seem significantly overclaimed.

**Questions:**

Could the authors provide a direct, quantitative comparison of the numerical error (e.g., MSE, L2 error) of their method against a standard baseline neural operator (like FNO) on the exact same problem instances, including compute time?

There is a pretty large literature on in-context learning, that explores the properties of LLM architectures for PDEs. While the approach is different, it seems natural to me to compare to these methods [1, 2, 3, 4].

Did you try to explore few-shot learning with parameter efficient fine tuning methods (e.g. LORA) to see how the LLM would behave, compared to a fully zero-shot setting?

[1] Chen et al, Data-Efficient Operator Learning via Unsupervised Pretraining and In-Context Learning. 2024
[2] Cao et al, VICON: Vision In-Context Operator Networks for Multi-Physics Fluid Dynamics Prediction. 2025.
[3] Serrano et al, Zebra: In-Context Generative Pretraining for Solving Parametric PDEs. 2025
[4] Kassaï Koupaï et al, ENMA: Tokenwise Autoregression for Generative Neural PDE Operators. 2025

**Details Of Ethics Concerns:**

No ethical concerns.

---

### Official Review · Reviewer_69VJ · 2025-10-30

**Soundness:** 1
**Presentation:** 2
**Contribution:** 2
**Rating:** 2
**Confidence:** 4

**Summary:**

The paper demonstrates that large language models can, to a certain extent, extrapolate the temporal dynamics of one-dimensional systems governed by partial differential equations. Using open-source LLM implementations (such as LLaMA 3) and a tokenization scheme that maps floating-point observations to three-digit integer tokens, the authors evaluate zero-shot performance on several equations, including the Allen–Cahn, Fisher–KPP, heat, and wave equations. The results indicate that LLMs are capable of short-term extrapolation and benefit from extended temporal context, but their predictions deteriorate rapidly over multiple time steps when compared to classical numerical solvers.

**Strengths:**

- Applying text-trained large language models to data governed by partial differential equations is an intriguing idea; nonetheless, it appears reasonable, given the existence of similar approaches on time series data.
- The tokenization scheme is simple and draws inspiration from the work of Gruver et al. [1].


[1] Gruver et al., Large language models are zero-shot time series forecasters., 2023

**Weaknesses:**

- I do not fully understand why the authors apply quantization to the inputs of the numerical solvers as well. Quantization is primarily a limitation of LLMs, not of traditional numerical solvers.

- As the authors note, LLMs do not perform zero-shot extrapolation on temporal PDEs—otherwise, increasing the number of spatial discretization points would similarly reduce the error, as seen in numerical solvers. A more modest claim would be that LLMs can extrapolate, to some extent, very simple spatiotemporal dynamics.

- The interpretation of Figure 5 is somewhat unclear. It appears to show that LLMs require a sufficient amount of input context to make confident predictions.

- The related work section omits prior studies on the application of LLMs to PDEs, specifically [1] and [2].

[1] Serrano et al., Zebra: In-context and Generative Pretraining for Solving Parametric PDEs, 2025.
[2] Zhou et al., Text2PDE: Latent Diffusion Models for Accessible Physics Simulation, 2024.

**Questions:**

- I am curious about the main motivation behind this work. Is the goal to establish a new benchmark for evaluating the advanced capabilities of large language models, or is it primarily aimed at advancing the field of PDE surrogates? I am inclined to think that the former interpretation would make more sense. However, the paper is not currently written in a way that clearly supports this. I would therefore suggest including a benchmark that compares more LLMs and uses a broader range of data, which should be feasible since no additional training is required.
- Do you believe that this approach could scale effectively to 2D datasets?

---

### Official Review · Reviewer_ATXj · 2025-10-31

**Soundness:** 2
**Presentation:** 2
**Contribution:** 1
**Rating:** 2
**Confidence:** 4

**Summary:**

This paper shows that a pretrained LLM, without any fine-tuning, can autoregressively continue solutions of time-dependent PDEs when the spatiotemporal field u(x,t) is linearly quantized to 3-digit integers and serialized as a comma-delimited spatial vector with semicolons separating time steps. Empirically, longer temporal context improves accuracy, and multi-step rollout errors grow with the horizon.

**Strengths:**

Extremely simple pipeline: the authors feed the serialized numeric sequence directly to a pretrained LLM with no fine-tuning and obtain next-step predictions autoregressively.

**Weaknesses:**

1. Lack of theoretical guarantees: PDE dynamics vary widely with governing equations, boundary, and initial conditions. In this zero-shot setup, the LLM receives no explicit PDE/BC/IC descriptors beyond the numeric prefix, so stability, conservation, and correctness are not guaranteed; the paper emphasizes analysis of emergent ICL behavior rather than proposing a principled solver.

2. Limited scope to 1D cases: although the study covers Allen–Cahn, Fisher–KPP, Heat, and Wave, all experiments are 1D; extending to 2D/3D complex PDEs (e.g., Navier–Stokes) remains open.

3. No explicit conditioning interface for BC/IC: apart from what can be inferred from the observed history, there is no mechanism to inject symbolic BC/IC or PDE parameters into the prompt.

**Questions:**

1. The authors state they don’t intend this as a PDE solver, yet the title sounds assertive. Would they consider softening the framing to avoid over-claiming? (e.g., emphasize “analysis of zero-shot ICL behavior.”)

2. Please evaluate complex boundary conditions and irregular grids to test robustness beyond the current 1D setups (they include Dirichlet and Neumann in 1D; broader cases would help).

3. Neural operators (FNO/DeepONet/FFNO, etc) can also do autoregressive PDE continuation. How does zero-shot LLM compare numerically (one-step and rollout RMSE/MAE) under the same discretization and context?

4. The PDE foundation-model line is cited, but a clearer positioning and experimental or analytical comparison would help: when and why is zero-shot LLM preferable to in-context operator learning or large PDE foundation models?

5. What happens if the codebook is enlarged (4 or 5 digits)?

---

### Note · Authors · 2025-11-15

**Comment:**

We thank the reviewers for their time and constructive feedback. Our motivation in this formulation of the work was not to propose using foundation models (such as Llama 3.1, Phi-4, and SmolLM 3) as PDE solvers, but rather to study the kinds of emergent in-context learning capabilities and shared behaviors these models exhibit when, without any natural-language prompting, PDE/BC/IC descriptors, or fine-tuning, they are exposed solely to the spatiotemporal trajectories of PDE dynamics. By using families of 1D linear and nonlinear PDE solutions (including Allen–Cahn, Fisher–KPP, Heat, and Wave equations) with randomized initial conditions as a clean and controlled setting, we aim to illuminate the numerical priors and inductive biases that arise purely from large-scale pretraining on textual data such as natural language and code. Across these PDEs, we uncover consistent in-context learning behavior in PDE continuation shared across different LLM families. We have decided to withdraw the submission and appreciate the reviewers’ suggestions and questions.

**Withdrawal Confirmation:**

I have read and agree with the venue's withdrawal policy on behalf of myself and my co-authors.